# Prevalence of depression in China during the early stage of the COVID-19 pandemic: a cross-sectional study in an online survey sample

Weina Liu ![ORCID],[1,2] Fengyun Yu ![ORCID],[1,3] Pascal Geldsetzer ![ORCID],[1,4] Juntao Yang ![ORCID],[5] Zhuoran Wang ![ORCID],[5] Todd Golden,[6] Lirui Jiao ![ORCID],[7] Qiushi Chen ![ORCID],[8] Haitao Liu ![ORCID],[5] Peixin Wu ![ORCID],[5] Chen Wang ![ORCID],[5,9] Till Bärnighausen ![ORCID],[10,11] Simiao Chen ![ORCID] [1,5]

WL, FY and PG are joint first authors.
TB and SC are joint senior authors.

**Correspondence to**
Dr Simiao Chen;
simiao.chen@uni-heidelberg.de

## ABSTRACT

**Objectives** We aimed to determine (1) the prevalence of depression during the COVID-19 pandemic among Chinese adults and (2) how depression prevalence varied by province and sociodemographic characteristics.

**Design** Cross-sectional study.

**Setting** National online survey in China.

**Participants** We conducted a cross-sectional online survey among adults registered with the survey company KuRunData from 8 May 2020 to 8 June 2020. We aimed to recruit 300–360 adults per province (n=14 493), with a similar distribution by sex and rural-urban residency as the general population within each of these provinces.

**Primary outcome** Participants completed the Patient Health Questionaire-9 (PHQ-9). We calculated the prevalence of depression (defined as a PHQ-9 score ≥10) nationally and separately for each province.

**Analysis** Covariate-unadjusted and covariate-adjusted logistic regression models were used to examine how the prevalence of depression varied by adults' sociodemographic characteristics. All analyses used survey sampling weights.

**Results** The survey was initiated by 14 493 participants, with 10 000 completing all survey questions and included in the analysis. The prevalence of depression in the national sample was 6.3% (95% CI 5.7% to 6.8%). A higher odds of depression was associated with living in an urban area (OR 1.50; 95% CI 1.18 to 1.90) and working as a nurse (OR 3.06; 95% CI 1.41 to 6.66). A lower odds of depression was associated with participants who had accurate knowledge of COVID-19 transmission prevention actions (OR 0.71; 95% CI 0.51 to 0.98), the knowledge that saliva is a main transmission route (OR 0.80; 95% CI 0.64 to 0.99) and awareness of COVID-19 symptoms (OR, 0.82; 95% CI 0.68 to 1.00).

**Conclusion** Around one in 20 adults in our online survey sample had a PHQ-9 score suggestive of depression. Interventions and policies to prevent and treat depression during the COVID-19 pandemic in China may be particularly needed for nurses and those living in urban areas.

## INTRODUCTION

As of middle January 2022, the WHO reports >326 million confirmed cases and >5.53 million deaths worldwide from the COVID-19 pandemic.[1] Effective vaccines are critical to

## Strengths and limitations of this study

► By sampling adults from each of China's provinces, this is the first study that is able to assess how the prevalence of depression during the COVID-19 pandemic varied across provinces.

► We explored how the prevalence of depression varied within provinces by individuals' socioeconomic characteristics, and are thus able to inform which population groups might be suitable target groups for prevention and treatment of depression during the pandemic.

► Our sample is unlikely to be representative of the general population of China because participants had to be registered with an online survey company (KuRunData) to be eligible for the survey.

► The study was conducted from May 8, 2020 to June 8, 2020, and thus our findings may not apply to other periods during the pandemic.

the containment of the COVID-19 pandemic and many vaccines are in clinical and preclinical development. As of middle January 2022, 139 vaccines are in clinical development, and 194 vaccines are in preclinical phases.[2] Moreover, almost 60% of the world population has received at least one dose of COVID-19 vaccine, and 9.68 billion doses have been administered worldwide, as well as 32.92 million doses are administered each day.[3] Although China is currently ranked as one of the countries with the most stringent restrictions, scoring the sixth highest in Stringency Index globally,[4] asymptomatic infection and mutated variants continue to pose an unprecedented threat. To reduce the impact of the pandemic, many countries imposed measures including city lockdowns, border closures, social isolation and quarantine. Long-term lockdowns and a large number of cases may affect not only social and economic

outcomes[5–10] but also disrupt normal life leading to pandemic-induced impairments in work and social functioning, including depression[11] and even suicide.[12]

Previous research suggests that mental health services are crucial to alleviate pandemic-related mental health impacts such as depression.[12] It is essential to understand the geographic distribution of depression to target mental health services on a national, regional, or local level. Non-pharmaceutical policies vary among countries, and they change dynamically as does the prevalence of depression. In China, the estimated overall prevalence of depression was 26.9%.[13] In all studies on depression during the initial COVID-19 pandemic, only four studies included the general population,[14–18] and none investigated the prevalence of depression by province. Moreover, several cross-sectional studies in different countries suggest an association between depression and sociodemographic characteristics such as female,[16] nurses,[15] low income,[15 19] young age,[14 16] lower education level,[20] having higher COVID-19 contraction risk, living in urban areas and social isolation.[21] However, some recent studies have contradicted these findings regarding the association between depression and these sociodemographic characteristics.[22] The prevalence of depression across China and the associated sociodemographic characteristics during the COVID-19 pandemic have yet to be fully understood.

Other factors, such as whether getting infected with COVID-19, or knowledge and perceptions about the virus, may also have an impact on psychological well-being and mental health. A previous study in India showed that inadequate knowledge of COVID-19 was associated with increased mental health problems.[23] Another study found the perception of COVID-19 may predict psychological well-being through the adoption of different types of coping strategies such as problem-focused coping and emotion-focused coping.[24] Increased knowledge regarding COVID-19 may contribute to better coping strategies, and hence positively affecting mental health.[25]

This study aimed to determine how the prevalence of depression varies by region, sociodemographic characteristics and other important factors that could affect psychological well-being and mental health, such as previous infections and knowledge and perceptions about the virus. These findings may help formulate effective interventions for improving mental health during different periods of the COVID-19 pandemic in different regions.

## METHODS
### Study design and sampling
We conducted a cross-sectional online survey from 8 May 2020 to 8 June 2020. The survey was carried out with Kurun Information Technology (KuRunData, Shanghai, China). The KurunData is an online internet panel for market research, which recruits participants and provides access to the questionnaire via Wechat mini. To assure the representativeness of the data, we used stratified sampling by sex, residence type and province. We calculated the size of each stratum based on a sequential method (online supplemental table S1) to reflect the province's population composition by sex and urban and rural residence (according to the China Statistical Yearbook 2020).[26] When sufficient participants were recruited for a given stratum, we halted recruitment for this stratum. Participants were paid ¥5 (US$0.77) for completing the questionnaire. All participants completed informed written consent. Finally, 14 493 participants initiated the online survey, with 10 000 participants completing the survey.

### Patient and public involvement
None of the participants was involved in the design of the questionnaire, nor the design, recruitment and implementation of the study. We plan to disseminate our study widely through the media, including in China.

### Measurements and covariates
As a part of investigation for willingness to participate in COVID-19 vaccination research, the questionnaire of our study included three parts: (1) COVID-19 awareness (perceived risk of death from COVID-19, knowledge of the transmission of COVID-19, awareness of recommended healthcare-seeking behaviour); (2) sociodemographic information (sex, age, highest education level, ethnicity, residence type, healthcare providers, annual household income, personal history of COVID-19 diagnosis, positive acquaintance COVID-19 diagnosis) and (3) the Patient Health Questionnaire-9 (PHQ-9) a depression screening tool.[27]

We used the PHQ-9 depression scale as a screening tool. There are several reasons for using this tool. At only nine items, it is suitable for a large-scale population survey and easily completed by the general population as a quick depression assessment. The nine diagnostic symptom criteria of the PHQ-9 correspond to the DSM-IV major depressive disorder criteria and can facilitate follow-up review of symptoms and the diagnosis process.[28] The PHQ-9 was also selected as it has good internal consistency,[29 30] reliability and validity,[31 32] with a Cronbach's alpha coefficient between 0.80 and 0.90. We defined a score of 10 or greater as depression; COVID-19 awareness consisted of nine items which were previously published.[33] Among these nine items, for categorical outcomes, data are expressed as binary or categorical (range: 0%–100%). For continuous outcomes, data are expressed as median (IQR).

### Statistical analysis
We excluded participants who answered less than half of the questions. All analyses applied sampling weights after excluding incomplete questionnaires. The sampling weights were calculated from the 2019 population census and the sampling quotas, accounting for some features of the survey, including oversampling for sex, residence type and province. Specifically, sampling weights are the inverse of the probability of selecting participants with some specific residence type (urban or rural), sex (male or

female), in some specfic province among the population. We used sampling weights adjusted for the survey design. For binary and categorical response options of knowledge about COVID-19, we calculated the percentage of participants with correct responses. For binomial proportions, we produced 95% CIs by the Wilson score interval. The prevalence of depression was stratified by participants' sociodemographic characteristics and overall knowledge about COVID-19. To examine how depression prevalence varied by sociodemographic characteristics, we used covariate-unadjusted and covariate-adjusted logistic regression with a binary indicator for each province (province-level fixed effects) and obtained OR. All analyses were conducted using R V.4.0.3 (R Foundation for Statistical Computing, Vienna, Austria).

## RESULTS
### Sociodemographic characteristics
The online survey was initiated by 14 493 participants, with 10 000 completing all survey questions and taking more than 2 min. A total of 4493 participants were excluded because they took less than 2 min to complete the survey, which we considered insufficient to read and answer all questions.

The majority of participants were male (50.8%), had at least a high school education (73.0%), lived in urban areas (59.4%) and had an annual household income ≥RMB90 000 (US$13 000) (51.6%). Han ethnicity accounted for the vast majority of participants (93.8%). Except for 9% of participants aged under 20 years, the percentage of other age groups ranged from 16.5% to 19.0%. Healthcare providers accounted for 4.1% of participants, with the largest category being community health workers (1.6%). 0.08% of participants had been diagnosed with COVID-19, and 0.34% of participants knew someone diagnosed with COVID-19. In addition, 6.0% of participants reported being previously diagnosed with depression. The rate of correct COVID-19 awareness ranged from 11.1% to 96.3%, knowledge of COVID-19 transmission prevention actions was lowest (11.1% (95% CI 10.5% to 11.7%)) and knowledge about the perceived risk of COVID-19 death among people with other diseases was highest (96.3% (95% CI 96.0% to 96.7%)) (see table 1.)

### Variation of depression prevalence by province
Overall, the depression prevalence was 6.3% (5.7%–6.8%). As seen in figure 1, the prevalence of depression in Henan was the highest (9.4% (6.6%–12.7%)) and in Hainan was the lowest (3.7% (1.9%–6.1%)). The prevalence of COVID-19 in Hubei province (1148 per million) tended to be higher than other areas in China, and Tibet (0.28 per million) saw the lowest prevalence (online supplemental table S2). While the prevalence of COVID-19 in other provinces tended to be similar, Hubei and Tibet were exceptions (figure 2).

### Variation of depression prevalence by sociodemographic characteristics within provinces
The results of covariate-unadjusted and covariate-adjusted logistic regression analysis of associated factors with depression are shown in table 2. After adjusting for covariates, urban residents and nurses had higher odds of depression than rural residents (OR 1.50; 95% CI 1.18 to 1.90) and other healthcare providers (OR 3.06; 95% CI 1.41 to 6.66). Participants who had correct knowledge of COVID-19 transmission prevention actions (OR 0.71; 95% CI 0.51 to 0.98) and knowledge of saliva as the main transmission route (OR 0.80; 95% CI 0.64 to 0.99) had lower odds of depression, as did participants who had accurate awareness of COVID-19 symptoms (OR 0.82; 95% CI 0.68 to 1.00).

For dichotomous outcomes, data are expressed as a percentage with the correct response (95% CI). For continuous outcomes, data are expressed as median (IQR).

## DISCUSSION
In our online survey sample, 6.3% of adults had depression as defined by a PHQ-9 ≥10. We found a higher prevalence of depression among certain population groups, including urban residents and nurses. Furthermore, we found that knowledge and awareness of COVID-19 were associated with a lower odds of depression. Under the strong assumption that these associations might be causal, this finding could indicate a 'protective effect' for mental health and thus indicate the importance of effective communication and education about COVID-19 amid the pandemic.

The correlation between urban residence and depression was positive in our study, which is consistent with several previous studies.[34 35] A possible interpretation of this finding is that while the virus could be transmitted more quickly in urban areas with a higher density population,[36] those in urban areas tend to have higher education levels and greater access to the latest updates on the COVID-19 pandemic.[34] Another explanation is that depression was more common in the urban than in rural areas in China before the COVID-19 pandemic. Moreover, the social distancing restrictions allowed less travel in cities than in rural areas, potentially contributing to the higher prevalence of depression in urban than in rural areas. This study also found that nurses had a higher odds of depression, consistent with previous studies.[37 38] One possible explanation was that as front-line healthcare workers, nurses had higher risk to be infected by longer contact with patients than doctors, and worked longer hours than usual,[39] which might make them become more frustrated.

Several studies have described the prevalence of depression during the COVID-19 pandemic in the general population. Among studies using the PHQ-9 scale with the same cut-off value (≥10), the prevalence of depression observed in our study (6.3%) was lower than a national

**Table 1** Sociodemographic characteristics, COVID-19 awareness and depression score (PHQ-9) (10 000 survey participants)

| Characteristic | Not depressed (%) (weighted; n=9396) | Depressed (%) (weighted; n=604) | Sample characteristics | | |
|---|---|---|---|---|---|
| | | | Proportion (weighted)* | N (%) (unweighted) | Proportion (unweighted)† |
| **Sex** | | | | | |
| Female | 49.1 | 46.2 | 48.9% | 4921 (49.2) | 48.9% |
| **Age group** | | | | | |
| <20 years | 10.5 | 11.0 | 10.5% | 900 (9.0) | 6.9% |
| 20–29 years | 17.1 | 17.9 | 17.1% | 1645 (16.5) | 20.8% |
| 30–39 years | 18.1 | 15.7 | 17.9% | 1895 (19.0) | 18.2% |
| 40–49 years | 18.7 | 20.5 | 18.9% | 1890 (18.9) | 22.1% |
| 50–59 years | 17.9 | 20.1 | 18.0% | 1820 (18.2) | 16.5% |
| ≥60 years | 17.7 | 14.9 | 17.5% | 1675 (16.8) | 15.4% |
| Highest education level | | | | | |
| Less than high school | 24.8 | 25.5 | 24.8% | 2792 (27.0) | 68.5% |
| High school/technical secondary school | 37.1 | 36.8 | 37.1% | 3733 (37.3) | 17.6% |
| College/undergraduate | 36.0 | 35.7 | 36.0% | 3369 (33.7) | 13.4% |
| Graduate school and above | 2.1 | 2.0 | 2.1% | 206 (2.0) | 0.6% |
| **Ethnicity** | | | | | |
| Han | 95.1 | 94.8 | 95.1% | 9381 (93.8) | 95.0% |
| Man | 0.4 | 0.6 | 0.5% | 149 (1.5) | 0.7% |
| Hui | 0.1 | 0.2 | 0.1% | 109 (1.1) | 0.8% |
| Zang | 1.8 | 1.5 | 1.8% | 103 (1.0) | 0.5% |
| Zhuang | 1.4 | 1.5 | 1.4% | 152 (1.5) | 1.2% |
| Other | 1.1 | 1.5 | 1.1% | 106 (1.1) | 1.8% |
| Province of residence | | | | | |
| Anhui | 4.6 | 4.0 | 4.5% | 360 (3.6) | 4.5% |
| Beijing | 1.5 | 1.9 | 1.5% | 360 (3.6) | 1.5% |
| Chongqing | 2.2 | 2.4 | 2.2% | 360 (3.6) | 2.2% |
| Fujian | 2.8 | 2.9 | 2.8% | 300 (3.0) | 2.8% |
| Gansu | 1.9 | 1.8 | 1.9% | 300 (3.0) | 1.9% |
| Guangdong | 8.1 | 9.5 | 8.2% | 360 (3.6) | 8.2% |
| Guangxi | 3.5 | 3.8 | 3.5% | 300 (3.0) | 3.5% |
| Guizhou | 2.6 | 1.8 | 2.6% | 300 (3.0) | 2.6% |
| Hainan | 0.7 | 0.4 | 0.7% | 300 (3.0) | 0.7% |
| Hebei | 5.4 | 5.5 | 5.4% | 360 (3.6) | 5.4% |
| Heilongjiang | 2.7 | 2.1 | 2.7% | 300 (3.0) | 2.7% |
| Henan | 6.6 | 10.3 | 6.9% | 360 (3.6) | 6.9% |
| Hubei | 4.2 | 4.3 | 4.2% | 360 (3.6) | 4.2% |
| Hunan | 4.9 | 4.7 | 4.9% | 300 (3.0) | 4.9% |
| Jiangsu | 5.7 | 7.1 | 5.7% | 360 (3.6) | 5.7% |
| Jiangxi | 3.4 | 2.8 | 3.3% | 300 (3.0) | 3.3% |
| Jilin | 1.9 | 1.8 | 1.9% | 300 (3.0) | 1.9% |
| Liaoning | 3.1 | 2.6 | 3.1% | 340 (3.4) | 3.1% |
| Neimengol | 1.9 | 1.2 | 1.8% | 300 (3.0) | 1.8% |
| Ningxia | 0.5 | 0.4 | 0.5% | 300 (3.0) | 0.5% |
| Qinghai | 0.4 | 0.3 | 0.4% | 300 (3.0) | 0.4% |
| Shaanxi | 2.8 | 2.6 | 2.7% | 360 (3.6) | 2.7% |

Continued

**Table 1** Continued

| Characteristic | Not depressed (%) (weighted; n=9396) | Depressed (%) (weighted; n=604) | Sample characteristics | | |
| --- | --- | --- | --- | --- | --- |
| | | | Proportion (weighted)* | N (%) (unweighted) | Proportion (unweighted)† |
| Shandong | 7.3 | 5.7 | 7.2% | 360 (3.6) | 7.2% |
| Shanghai | 1.7 | 1.4 | 1.7% | 300 (3.0) | 1.7% |
| Shanxi | 2.7 | 1.7 | 2.8% | 300 (3.0) | 2.8% |
| Sichuan | 5.9 | 7.1 | 6.0% | 360 (3.6) | 6.0% |
| Tianjin | 1.1 | 1.0 | 1.1% | 360 (3.6) | 1.1% |
| Tibet | 0.2 | 0.3 | 0.3% | 300 (3.0) | 0.3% |
| Xinjiang | 1.8 | 2.2 | 1.8% | 300 (3.0) | 1.8% |
| Yunnan | 3.5 | 2.7 | 3.5% | 300 (3.0) | 3.5% |
| Zhejiang | 4.2 | 3.5 | 4.2% | 360 (3.6%) | 4.2% |
| Residence type | | | | | |
| Urban | 60.9 | 62.9 | 61.1% | 5935 (59.4%) | 60.6% |
| Healthcare providers | | | | | |
| No | 96.2 | 93.6 | 96.0% | 9597 (96.0%) | 99.0% |
| Nurse | 0.4 | 1.3 | 0.5% | 55 (0.6%) | 0.3% |
| Physician | 0.8 | 1.3 | 0.9% | 84 (0.8%) | 0.5% |
| Community health worker | 1.4 | 2.5 | 1.5% | 157 (1.6%) | <0.1% |
| Pharmacist | 0.1 | 0.1 | 0.1% | 17 (0.2) | <0.1% |
| Other healthcare provider | 0.9 | 1.2 | 1.0% | 90 (0.9%) | 0.1% |
| Annual household income (RMB) | | | | | |
| <RMB30 000 | 5.7 | 6.5 | 5.8% | 560 (5.6%) | |
| RMB30 000–RMB59 999 | 15.0 | 13.9 | 14.9% | 1670 (16.7%) | – |
| RMB60 000–RMB89 999 | 21.7 | 19.6 | 21.6% | 2303 (23.0%) | – |
| RMB90 000–RMB119 999 | 25.9 | 30.9 | 26.2% | 2704 (24.0%) | – |
| RMB120 000–RMB149 999 | 14.2 | 14.9 | 14.2% | 1211 (12.1%) | – |
| RMB150 000–RMB199 999 | 11.1 | 8.8 | 11.1% | 974 (9.7%) | – |
| ≥RMB200 000 | 6.4 | 5.4 | 6.4% | 578 (5.8%) | – |
| Personal COVID-19 diagnosis history | | | | | |
| No | 99.9 | 99.8 | 99.9% | 9992 (99.9%) | |
| Yes | 0.1 | 0.2 | 0.1% | 8 (0.08%) | |
| Postive acquaintance COVID-19 diagnosis | | | | | |
| Family member | 0.1 | 0.2 | 0.1% | 5 (0.05%) | |
| Friend | 0.2 | 0.0 | 0.1% | 12 (0.12%) | |
| Neighbour | 0.1 | 0.0 | 0.1% | 7 (0.07%) | |
| Cowoker | 0.1 | 0.3 | 0.1% | 10 (0.1%) | |
| Other | 0.0 | 0.0 | 0.0% | 1 (0) | |
| COVID-19 awareness | | | | | |
| Perceived risk of death among vulnerable groups | 3.3 (IQR: 1.0–4.0) | 3.5 (IQR: 1.0–4.0) | 3.3% (IQR: 1.0%–4.0%) | 3.3% (IQR: 1.0%–4.0%) | |
| Perceived risk of death among people with other diseases | 96.6% (95% CI 96.2% to 96.9%) | 94.4% (95% CI 92.2% to 96.0%) | 96.4% (95% CI 96.0% to 96.8%) | 96.3 (95% CI 96.0% to 96.7%) | |
| Perceived elderly as a high-risk group of transmission | 92.6% (95% CI 92.0% to 93.1%) | 92.6% (95% CI 90.1% to 94.4%) | 92.6% (95% CI 92.0% to 93.1%) | 92.7% (95% CI 92.2% to 93.2%) | |
| Awareness of vaccine availability protecting against transmission | 76.7% (95% CI 75.8% to 77.5%) | 74.9% (95% CI 71.2% to 78.1%) | 76.5% (95% CI 75.7% to 77.3%) | 76.0% (95% CI 75.1% to 76.8%) | |

Continued

**Table 1** Continued

| Characteristic | Not depressed (%) (weighted; n=9396) | Depressed (%) (weighted; n=604) | Sample characteristics | | Proportion (unweighted)† |
| --- | --- | --- | --- | --- | --- |
| | | | Proportion (weighted)* | N (%) (unweighted) | |
| Awareness that masks are highly effective in protecting against transmission | 89.3% (95% CI 88.7% to 89.9%) | 88.%0 (95% CI 85.1% to 90.4%) | 89.2% (95% CI 88.6% to 89.8%) | 89.8% (95% CI 89.2% to 90.4%) | |
| Knowledge of transmission prevention actions | 11.6% (95% CI 10.9% to 12.2%) | 8.1% (95% CI 6.2% to 10.6%) | 11.4% (95% CI 10.8% to 12.0%) | 11.1% (95% CI 10.5% to 11.7%) | |
| Knowledge of saliva as main transmission route | 82.6% (95% CI 81.8% to 83.3%) | 76.9% (95% CI 73.4%to 80.1%) | 82.2% (95% CI 81.5% to 83.0%) | 82.1% (95% CI 81.3% to 82.8%) | |
| Awareness of COVID-19 symptoms | 69.4% (95% CI 68.5% to 70.3%) | 64.4% (95% CI 60.5% to 68.1%) | 69.1% (95% CI 68.2% to 70.0%) | 67.1% (95% CI 66.1% to 68.0%) | |
| Awareness of recommended healthcare-seeking behaviour | 71.6% (95% CI 70.7% to 72.5%) | 68.2% (95% CI 64.3% to 71.8%) | 71.4% (95% CI 70.5% to 72.3%) | 68.8% (95 CI 67.9% to 69.7%) | |

*Weighted using survey sampling weights.
†As per the 2020 China Statistical Yearbook.
PHQ-9, Patient Health Questionaire-9.

study among 56 679 participants conducted 28 February 2020 to 11 March 2020 in China (10.8%)[40] as well as a study of 1470 individuals among the general population during the COVID-19 outbreak in the USA from 31 March 2020 to 13 April 2020 (27.8%).[41] Most of the other studies were conducted in February 2020—at the peak of the COVID-19 epidemic, and the prevalence of depression ranged from 8.3% to 48.3%.[40 42–44] One possible explanation is that our study occurred in May when the epidemic had decreased in severity, and the prevalence of depression may have diminished. Another possible reason for a lower depression prevalence found in our study may be that with the continuous strict quarantine policy, the government provided timely mental health service in response to COVID-19.[45]

The prevalence of COVID-19 in 31 provinces tended to be similar except for Hubei, which had the highest prevalence (figure 2). However, the proportion of the population by province with depression (PHQ-9≥10) did not

seem to be significantly associated with the prevalence of COVID-19 cases. Moreover, from the covariate-adjusted logistic regression results, we found that depression was not associated with the prevalence of COVID-19 cases confirmed by province, which also suggested that the prevalence of depression is not associated with the severity of the regional epidemic during the COVID-19 outbreak. One explanation is that resilience plays a protective role in mitigating the impact of stress and trauma on depressive symptoms and the consequences associated with depressive symptoms.[46] During the initial phase of the COVID-19 outbreak in China, the Chinese government enforced a rigid social distancing policy through social media and strongly promoted hand washing, surface disinfection and the use of protective masks.[47] In Wuhan, the most affected city, people with mild and asymptomatic infection received care in Fangcang shelter hospitals, which are designed for facility-based isolation, treatment and monitoring,[47] and have been proven to be an effective

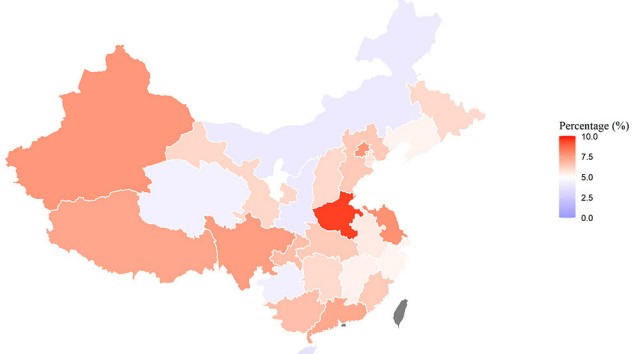

**Figure 1** Proportion of the population reporting depression (Patient Health Questionaire-9 ≥10) by province* (10 000 participants, 8 May 2020–8 June 2020). *Taiwan, Hong Kong and Macao are shown in grey. Source of map: http://datav.aliyun.com/portal/school/atlas/area_selector.

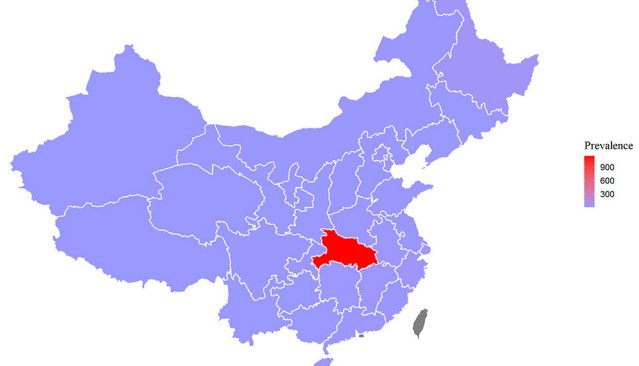

**Figure 2** Prevalence of COVID-19 by province* (1 January 2020–8 June 2020). *Taiwan, Hong Kong and Macao are shown in grey. Source of map: http://datav.aliyun.com/portal/school/atlas/area_selector.

**Table 2** Covariate-unadjusted and covariate-adjusted logistic regressions of depression onto sociodemographic characteristics

| | Covariate-unadjusted OR (95% CI)† | Covariate-adjusted OR (95% CI)‡ | Depression prevalence (95% CI) |
|---|---|---|---|
| Overall | | | 604 (6.3%) (5.7% to 6.8%) |
| **Sex** | | | |
| Male | Ref. | Ref. | 323 (6.6%) (5.8% to 7.4%) |
| Female | 0.90 (0.75 to1.07) | 0.89 (0.74 to 1.07) | 281 (5.9%) (5.2% to 6.7%) |
| **Age** | | | |
| 18–19 years | Ref. | Ref. | 65 (6.5%) (4.9% to 8.4%) |
| 20–29 years | 1.01 (0.72 to 1.42) | 1.06 (0.72 to 1.56) | 117 (6.5%) (5.2% to 8.0%) |
| 30–39 years | 0.85 (0.59 to 1.21) | 0.91 (0.60 to 1.37) | 98 (5.5%) (4.3% to 6.8%) |
| 40–49 years | 1.07 (0.76 to 1.49) | 1.12 (0.77 to 1.63) | 114 (6.8%) (5.5% to 8.3%) |
| 50–59 years | 1.09 (0.78 to 1.53) | 1.05 (0.72 to 1.54) | 120 (7.0%) (5.7% to 8.4%) |
| >60 years | 0.83 (0.58 to 1.19) | 0.82 (0.52 to 1.31) | 90 (5.3%) (4.2% to 6.6%) |
| **Highest education level** | | | |
| Less than high school | Ref | Ref | 165 (7.2%) (5.8% to 8.7%) |
| High school/technical secondary school | 1.48 (0.85 to 2.59) | 1.16 (0.61 to 2.21) | 225 (6.2%) (5.3% to 7.2%) |
| College/undergraduate | 1.45 (0.83 to 2.54) | 1.12 (0.58 to 2.14) | 202 (6.2%) (5.3% to 7.2%) |
| Graduate school and above | 1.37 (0.58 to 3.25) | 1.09 (0.43 to 2.75) | 12 (5.9%) (2.6% to 10.5%) |
| **Residence type** | | | |
| Rural | Ref. | Ref. | 243 (6.0%) (5.1% to 6.9%) |
| Urban | 1.09 (0.90 to 1.31) | 1.50 (1.18 to 1.90)** | 361 (6.5%) (5.8% to 7.2%) |
| **Healthcare providers** | | | |
| No | Ref. | Ref. | 564 (6.1%) (5.6% to 6.7%) |
| Nurse | 2.61 (1.18 to 5.75)* | 3.06 (1.41 to 6.66)** | 7 (16.1%) (5.1% to 31.5%) |
| Physician | 1.48 (0.65 to 3.38) | 1.51 (0.67 to 3.40) | 7 (9.2%) (2.9% to18.5%) |
| Community health worker | 1.66 (0.97 to 2.86) | 1.62 (0.94 to 2.78) | 16 (10.4%) (5.5% to 16.6%) |
| Pharmacist | 0.91 (0.18 to 4.71) | 1.17 (0.24 to 5.69) | 2 (11.8%) (NA~NA) |
| Other healthcare provider | 1.38 (0.65 to 2.92) | 1.31 (0.60 to 2.90) | 8 (8.1%) (2.9% to 15.5%) |
| **Annual household income (RMB)** | | | |
| <RMB30 000 | Ref. | Ref. | 39 (7.1%) (4.8% to 9.8%) |
| RMB30 000–RMB59 999 | 0.87 (0.58 to 1.31) | 0.92 (0.60 to 1.42) | 98 (5.8%) (4.6% to 7.2%) |
| RMB60 000–RMB89 999 | 0.82 (0.55 to 1.22) | 0.87 (0.56 to 1.34) | 132 (5.7%) (4.7% to 6.9%) |
| RMB90 000–RMB119 999 | 1.05 (0.71 to 1.56) | 1.14 (0.74 to 1.75) | 182 (7.4%) (6.3% to 8.7%) |
| RMB120 000–RMB149 999 | 0.87 (0.56 to 1.34) | 0.92 (0.56 to 1.49) | 78 (6.6%) (5.1% to 8.2%) |
| RMB150 000–RMB199 999 | 0.68 (0.42 to 1.10) | 0.70 (0.41 to 1.20) | 44 (5.0%) (3.6% to 6.7%) |
| ≥RMB200 000 | 0.69 (0.40 to 1.20) | 0.72 (0.39 to 1.32) | 31 (5.3%) (3.4% to 7.7%) |
| **COVID-19 awareness** | | | |
| Perceived risk of death among vulnerable groups | 1.02 (0.99 to 1.05) | 1.02 (0.99 to 1.05) | NA |
| Perceived risk of death among people with other diseases | 0.87 (0.72 to 1.05) | 0.88 (0.73 to 1.07) | 395 (6.0%) (5.4% to 6.7%) |
| Perceived elderly as a high-risk group of transmission | 1.00 (0.71 to 1.40) | 1.05 (0.75 to 1.47) | 558 (6.3%) (5.7% to 6.9%) |
| Awareness of vaccine availability protecting against transmission | 0.92 (0.75 to 1.13) | 0.91 (0.73 to 1.13) | 447 (6.1%) (5.5% to 6.8%) |
| Awareness that masks are highly effective in protecting against transmission | 0.88 (0.67 to 1.16) | 0.89 (0.68 to 1.17) | 533 (6.2%) (5.6% to 6.8%) |
| Knowledge of transmission prevention actions | 0.68 (0.49 to 0.94)* | 0.71 (0.51 to 0.98)* | 50 (4.5%) (3.2% to 6.0%) |
| Knowledge of saliva as main transmission route | 0.74 (0.60 to 0.91)** | 0.80 (0.64 to 0.99)* | 464 (5.9%) (5.3% to 6.5%) |
| Awareness of COVID-19 symptoms | 0.81 (0.67 to 0.97)* | 0.82 (0.68 to 1.00)* | 386 (5.8%) (5.2% to 6.5%) |

Continued

**Table 2** Continued

| | Covariate-unadjusted OR (95% CI)† | Covariate-adjusted OR (95% CI)‡ | Depression prevalence (95% CI) |
|---|---|---|---|
| Awareness of recommended healthcare-seeking behaviour | 0.84 (0.69 to 1.02) | 0.81 (0.65 to 1.01) | 404 (6.0%) (5.4% to 6.7%) |
| Postive acquaintance COVID-19 diagnosis | | | |
| No | Ref. | Ref. | 602 (6.3%) (5.7% to 6.8%) |
| Yes | 1.60 (0.44 to 5.84) | 1.22 (0.32 to 4.61) | 2 (10.0%) (0.5% to 29.5%) |
| COVID-19 cases confirmed by the province | | | |
| Prevalence of COVID-19 by province | 1.00 (0.99 to 1.01) | 1.00 (0.99 to 1.02) | NA |

*represents p<0.05; **represents p<0.01; ***represents p<0.001
†Covariate-unadjusted logistic regression results (All regressions included each one of the variables shown in the table and adjusted for each province (province-level fixed effects)).
‡Covariate-adjusted logistic regression results (All regressions included all variables shown in the table and adjusted for each province (province-level fixed effects)).
NA, not available.

method to control the epidemic.[48 49] Fangcang shelter hospitals also provided mental health counselling services and social support to help patients recover during isolation,[50] which may also reduce the incidence of depression in hard-hit areas. Due to very limited confirmed cases in our study, it was difficult to investigate the association between depression and the number of confirmed cases (0.1% participants diagnosed with COVID-19, 0.48% participants with an acquaintance with confirmed COVID-19).

This study, for the first time, provides preliminary evidence that knowledge of COVID-19 transmission prevention actions, knowledge of saliva as main transmission route and awareness of COVID-19 symptoms may be associated with a lower risk of depression. Compared with previous research,[34 51 52] our study reinforced the positive association between precautionary measures as well as awareness of COVID-19 symptoms and depression. One possible explanation could be that accurate knowledge of COVID-19 transmission prevention actions and awareness of COVID-19 symptoms may help alleviate the stress caused by fear of COVID-19 pandemic. Stress, as a risk factor for depression, plays a role in triggering depression by biological mechanisms such as hypothalamic-pituitary-adrenal (HPA) axis stress response processes and hormonal and neurotransmitter systems.[53] Therefore, accurate knowledge and perceptions of COVID-19 could indirectly affect depression through the reduction in stress. This finding suggests that, to reduce the prevalence of depression, effective communication and education of COVID-19 preventive measures and recommended healthcare-seeking behaviours are urgently needed.

### Limitations

This is the first study that investigated the prevalence of depression by province during the early stage of the COVID-19 pandemic in China; however, our study has several limitations. First, although our sample size was large, all participants needed to register with KuRunData and therefore our sample is unlikely to be representative of the general population of China. As this was an online survey, participants had access to a computer or smartphone and therefore tended to be younger and highly educated. Although we used stratified sampling to increase the representativeness of the data, it is still difficult to avoid response bias as potential participants with depression might be either less or more interested in taking part in the survey. Also, our measure of depression was the PHQ-9 score, which is a tool for screening for depression but is not recommended for a clinical diagnosis of the condition.

### CONCLUSIONS

In conclusion, the prevalence of depression in our national online survey sample was around 6% during the initial stage of the COVID-19 pandemic. Depression was more common among nurses and in urban areas. Accurate knowledge of COVID-19 transmission and awareness of COVID-19 symptoms were associated with lower odds of depression. These findings may assist targeted efforts to prevent or treat depression in China to those most in need of mental health services during the COVID-19 pandemic.

**Author affiliations**
[1]Heidelberg Institute of Global Health, Heidelberg University, Heidelberg, Germany
[2]Children and Adolescents Health Promotion, Jiangsu Provincial CDC, Nanjing, Jiangsu, China
[3]Department of Industrial Engineering, Tsinghua University, Beijing, China
[4]Division of Primary Care and Population Health,Department of Medicine, Stanford University, Stanford, California, USA
[5]Chinese Academy of Medical Sciences & Peking Union Medical College, Beijing, China
[6]National Cancer Institute Division of Cancer Control and Population Sciences, Rockville, Maryland, USA
[7]Mailman School of Public Health, Columbia University, New York City, New York, USA
[8]Harold and Inge Marcus Department of Industrial and Manufacturing Engineering, The Pennsylvania State University, University Park, Pennsylvania, USA
[9]Department of Pulmonary and Critical Care Medicine of Respiratory Medicine, China-Japan Friendship Hospital, Beijing, China

[10]Institute of Public Health, Universitatsklinikum Heidelberg Institut fur Global Health, Heidelberg, Germany

[11]Department of Global Health and Population, Harvard T.H. Chan School of Public Health, Boston, Massachusetts, USA

**Contributors** WL and SC conceived the idea, JY, ZW, HL, PW collected data, FY and WL analysed and verified data, WL wrote the first draft, PG and TG edited the manuscript, LJ, QC, SC, CW and TB collaboratively reviewed the manuscript. SC acted as guarantor for the overall content of the article.

**Funding** This work was supported by the Bill & Melinda Gates Foundation (Project INV-006261) and the Sino-German Center for Research Promotion (Project C-0048). The funding source had no involvement in study design and data collection.

**Map disclaimer** The inclusion of any map (including the depiction of any boundaries therein), or of any geographic or locational reference, does not imply the expression of any opinion whatsoever on the part of BMJ concerning the legal status of any country, territory, jurisdiction or area or of its authorities. Any such expression remains solely that of the relevant source and is not endorsed by BMJ. Maps are provided without any warranty of any kind, either express or implied.

**Competing interests** None declared.

**Patient and public involvement statement** None of the participants were involved in the design of the questionnaire, nor the design, recruitment, and implementation of the study.

**Patient consent for publication** Not required.

**Ethics approval** This research was approved by the institutional review board of Chinese Academy of Medical Sciences and Peking Union Medical College(CAMS&PUMC-IEC-2020–001), and authors had access only to deidentified data.

**Provenance and peer review** Not commissioned; externally peer reviewed.

**Data availability statement** Data are available on reasonable request.

**ORCID iDs**
Weina Liu http://orcid.org/0000-0001-5377-9714
Fengyun Yu http://orcid.org/0000-0002-5607-1102
Pascal Geldsetzer http://orcid.org/0000-0002-8878-5505
Juntao Yang http://orcid.org/0000-0003-1180-391X
Zhuoran Wang http://orcid.org/0000-0001-5175-8645
Lirui Jiao http://orcid.org/0000-0002-2505-7447
Qiushi Chen http://orcid.org/0000-0003-4031-2669
Haitao Liu http://orcid.org/0000-0002-6184-1466
Peixin Wu http://orcid.org/0000-0002-1288-0610
Chen Wang http://orcid.org/0000-0001-7857-5435
Till Bärnighausen http://orcid.org/0000-0002-4182-4212
Simiao Chen http://orcid.org/0000-0003-0555-2157

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
