## [Reviewer comments · BMJ Open]

ARTICLE DETAILS

TITLE (PROVISIONAL)	The prevalence of depression in China during the early stage of the COVID-19 pandemic: A cross-sectional study in an online survey sample
AUTHORS	Liu, Weina; Yu, Fengyun; Geldsetzer, Pascal; Yang, Juntao; Wang, Zhuoran; Golden, Todd; Jiao, Lirui; Chen, Qiushi; Liu, Haitao; Wu, Peixin; Wang, Chen; Bärnighausen, Till; Chen, Simiao

VERSION 1 – REVIEW

REVIEWER	Chen, Min The Third Affiliated Hospital of Guangzhou Medical University, 1. Department of Fetal Medicine and Prenatal Diagnosis
REVIEW RETURNED	24-Oct-2021

GENERAL COMMENTS	bmjopen-2021-056667 Depression in China during the coronavirus 2019 pandemic: A cross-sectional study Comments: Thanks for the opportunity of reviewing this interesting and original paper. This manuscript is well written, but there were some limitations. And English service is needed. Introduction Page 5 line 6 “As of June 2021”, do you mean early, middle or late June 2021? Ps check the citation for consistence. Page 5 line 52-55 “More recently, literature has emerged that offers contradictory findings about the association between depression and these sociodemographic characteristics 17 19” Ps rewrite this sentence for clarification. Study Design and Sampling Have you consult a statistician whether the use of sampling is appropriate? Is your sample representative of your population? Ps attach your Institute Ethics approval for this study. Page 7 line 47-48 What is the rationale to use the Patient Health Questionnaire-9 (PHQ-9)? Why not use other Depression scale?
--

	Ps comment in the discussion session and cite this as reference. https://www.ncbi.nlm.nih.gov/books/NBK349029/table/ch1.t2/ Measurements Since data on knowledge of COVID-19 was published, this part can be shortened. Statistical Analysis Page 8 last sentence. I suggest rephrasing as “All analyses were conducted using R version 4.0.3 (R Foundation for Statistical Computing, Vienna, Austria).” Results Page 9 line 17-18 Write out number 50.8 in words: Fifty-point eight percent. Do not start a sentence with Number. Discussion Your sample is unlikely to be representative of the general population of China. So it is not a well-designed epidemiological study, is the term “prevalence of depression” appropriate? Depression prevalence shall be replaced by “incidence of depression”. Why nurses are at high-risk of depression? Ps elaborate? This is the first study that investigated the prevalence of depression by province during the initial stage of the COVID-19 pandemic in China? I suggest a literature search and a literature or systematic review, or a mini-review with illustrative table including this study could add more value to the work. References. I suggest updating the references, as several papers have now been published on depression amongst pregnant women during COVID-19. The reference style is needed to be changed, family name first. for example : 14 36 17 and 19 are duplicated. Ps check. Table 1. Education Level shall be converted to western style so that readers from different country can understand.
--	---

REVIEWER	Wang, Weichen The University of Hong Kong
REVIEW RETURNED	29-Oct-2021

GENERAL COMMENTS	The paper studies the association of some important potential factors with the prevalence of depression in China during the COVID-19 pandemic time period. Specifically, the factors under
--

consideration include sociodemographic characteristics, whether one works as certain type of health care providers, COVID-19 awareness, and acquaintance with COVID-19 positive cases. The work claims to be the first to address how the prevalence of depression varies by regions, sociodemographic characteristics, and other things that affect mental health in China.

For the study design, the authors collaborated with an information technology company in China and conducted a cross-section survey in 2020 and sampled 10,000 adults from 31 provinces. The COVID-19 awareness and the depression level, together with personal demographic information, were measured from a questionnaire. A Patient Health Questionnaire-9 (PHQ-9) score larger than 4 was considered depression. The COVID-19 awareness consisted of questions such as perceived risk, COVID-19 symptoms, awareness of vaccine availability, knowledge of transmission prevention actions etc.

Through the study, the authors found that (1) depression was more common among urban residents, nurses and those who has a family member diagnosed with COVID-19; (2) knowledge of COVID-19 transmission and healthcare-seeking behavior can effectively reduce the risk of depression. I think the paper provides intuitive conclusions. Through the data analysis, we get more support and details on the effect of different factors under consideration on depression.

I only have some minor comments listed below.

Minor remarks:

1. Page 7, line 20, "yuan".
2. Page 7, line 48, "a score 4". Why do we use 4 as the threshold? Is it for a more balanced data in logistic regression? Or do we have literature convention or argument from mental health studies? In some other places of the paper, the definition becomes > 4 (e.g. Page 9, line 48). In Table 1, no depression is defined as PHQ-9 < 4 and depression is defined as PHQ-9 ≥ 4 . Please be consistent with how to handle the case with PHQ-9 = 4.
3. Page 8, line 36, "using χ^2 to compare". I do not follow what the authors mean here. Please elaborate.
4. Page 8, line 43, "an odds ratio" "odds ratios".
5. The paper has no clear reference to Table 1. Please add a sentence in the main text about something like "Please refer to Table 1 on summary statistics of ...".

	6. Page 15, lines 27-34. I do not follow the sentence. That the prevalence of depression had not been alleviated may explain why the prevalence was still high. How can we explain A by A itself? 7. Page 17, line 6, "effective communication" ! "effective communication" 8. Page 17, lines 13-22. People in urban areas have higher education and greater access to COVID information. How come their depression is higher or positively correlated with urban dummy variable? The authors argued more knowledge on COVID should reduce the depression. They seem to contradict here. Is it possible that the higher depression may not be explained as something related with COVID, but just by the fact that working in a city is more stressful? 9. Page 17, line 31, "more vulnerable" ! "are more vulnerable". 10. Page 17, line 34, "contracting" ! "contacting". 11. Page 17, line 45, "initial stage"? The data was collected in May 2020. Is this still considered as initial stage? By this time, the peak outbreak of the pandemic in China has passed. 12. Page 18, lines 4-6. The study design was cross sectional, so the authors "cannot make temporal conclusions". But the paper does compare the prevalence of depression with previous studies in lines 10-18, Page 15. Could you explain what the meaning of "temporal conclusions" is here? 13. Page 18, line 6, "cannot" ! "cannot". 14. Page 18, line 18, "was associated" ! "were associated".
--	---

REVIEWER	Daly, Michael National University of Ireland
REVIEW RETURNED	03-Nov-2021

GENERAL COMMENTS	This study includes a large (10,000) sample from across each of China's provinces and participants completed a well validated depression screen. The paper is well-written, includes reference to relevant research, and analyses incorporate survey weights and examine important predictors. Introduction 1) Please update COVID-19 prevalence and mortality figures (these are from June, 2021).
--

2) Update description of vaccines (being developed)

3) Description of China having returned to 'normal life' whereas other countries have not does not tally with objective indicators such as the COVID-19 Stringency Index where China is currently ranked as one of the countries with the most stringent restrictions globally (<https://ourworldindata.org/grapher/covid-stringency-index>). If China had returned fully to normal life it would also beg the question as to why it is necessary to conduct this study. Please update the introduction to reflect how China compares to other nations on indicators such as the Stringency Index.

Method:

4) Please include more information on how participants were recruited / why people register with KuRunData (e.g. is this an existing online internet panel for market research etc.).

5) The typical cut-off used on the PHQ-9 is greater than or equal to 10. A cut-off of greater than or equal to 4 is unusual. Even mild depressive symptoms are typically gauged using a score of greater than or equal to 5 on this measure. Unless there is a strong rationale not to do so I suggest using the more conventional cut-off of ≥ 10 which the original Kroenke et al (2001) paper has shown to have a sensitivity of 88% and a specificity of 88% for major depression. The use of the 4 cut-off partly explains why the rate of depression appears to be very high in the current results.

Results

6) Please provide more information on how the sampling weights were calculated. The method section indicates that "The sampling weights were calculated by the inverse of the probability of age, residence type, and sex of selecting participants." Yet, looking at Table 1 the weights appear to produce an alignment with the China 2020 Statistical Yearbook based on Sex and Province of Residence. What role did age and residence type play in the weighting model?

7) A table such as Table 1 including the weighted prevalence of depression by sociodemographic characteristics would be worth including.

8) It is not clear how you are finding such large odds ratios for family and neighbor confirmed COVID ("and having a family member (OR, 23.75; 95% CI, 1.86-303.51) or neighbor (OR, 19.71; 95% CI, 1.54 - 252.22)"). Can you give details of the prevalence of depression in these groups and demonstrate how the ORs are accurate? I would suggest combining these groups as they are very low N which may be producing these extreme ORs (e.g. Any Positive Acquaintance COVID-19 Diagnosis would be 35 of 10,000 participants or 0.35%).

Discussion:

9) The discussion begins "This is the first study that quantified the prevalence of depression symptoms and its variation by province and sociodemographic characteristics in China". Where in the paper was the prevalence of depression quantified by province?

	10) The PHQ-9 cut-off described in the method was ≥ 4 and this was reported as greater than 4 in the discussion. Please provide the correct description. 11) Discussion of depression rates identified in other studies needs to be better contextualised. For instance, are there other nationally representative studies that have used the PHQ-9 during COVID-19 and reported on the same cut-offs? The rate of depression identified will depend on the measure and cut-off used which needs to be noted.
--	---

VERSION 1 – AUTHOR RESPONSE

Responses to Reviewers

Response to Reviewer 1

We thank Reviewer 1 for the valuable and constructive comments. We have carefully incorporated the reviewer's advice and as a result our paper has improved substantially.

Below, we explain in detail how we have taken each of the comments into account in the revision of our manuscript. We show the reviewer's comments in italics and reply to them in standard font.

[Comment-1] *Page 5 line 6 "As of June 2021", do you mean early, middle or late June 2021?*

Ps check the citation for consistence.

[Answer] Thank you very much for this comment. For this comment, we carried out modifications on the manuscript, as shown below:

(1) The authors added the word "middle" into the sentence and updated the reference (**Page 5, paragraph 1**):

"As of middle January 2022 the World Health Organization (WHO) reports >326 million confirmed cases and >5.53 million deaths worldwide from the coronavirus disease 2019 (COVID-19) pandemic²"

[Comment-2] *Study Design and Sampling*

Have you consult a statistician whether the use of sampling is appropriate?

Is your sample representative of your population?

Ps attach your Institute Ethics approval for this study.

[Answer] Thank you for this comment. The authors consulted a statistician and added the following text to the Methods section. (**Page 7, paragraph 1**). Because this study is a part of investigation for willingness to participate in COVID-19 vaccination research, the authors attached the Ethics approval for willingness to participate in COVID-19 vaccination research in upload file.

"The KurunData is an online internet panel for market research, which recruits participants and provides access to the questionnaire via Wechat mini. To assure the representativeness of the data, we utilized stratified sampling by sex, residence type and province. We calculated the size of each stratum based on a sequential method (Table S1) to reflect the province's population composition by sex and urban

and rural residence (according to the China Statistical Yearbook 2020¹). When sufficient participants were recruited for a given stratum, we halted recruitment for this stratum. Participants were paid 5 yuan (US\$0.77) for completing the questionnaire. All participants completed informed written consent. Finally, 14493 participants initiated the online survey, with 10000 participants completing the survey.”

\

[Comment-3] *Page 7 line 47-48*

What is the rationale to use the Patient Health Questionnaire-9 (PHQ-9)?

Why not use other Depression scale? Ps comment in the discussion session and cite this as reference. <https://www.ncbi.nlm.nih.gov/books/NBK349029/table/ch1.t2/>

[Answer] The rationale for using the Patient Health Questionnaire-9 (PHQ-9) was added to the Discussion section and the suggested reference cited. (**Page 8, paragraph 1**)

“We utilized the PHQ-9 depression scale as a screening tool. There are several reasons for using this tool. At only nine items, it is suitable for a large-scale population survey and easy to complete by the general population as a quick depression assessment. The nine diagnostic symptom criteria of the PHQ-9 correspond to the DSM-IV major depressive disorder (MDD) criteria and can facilitate follow-up review of symptoms and the diagnosis process³. The PHQ-9 was also selected as it has good internal consistency^{4,5}, reliability, and validity^{6,7} with a Cronbach’s alpha coefficient between 0.80 and 0.90.”

[Comment-4] *Measurements*

Since data on knowledge of COVID-19 was published, this part can be shortened.

[Answer] The authors shortened the description of COVID-19 awareness published previously. (**Page 7, paragraph 3**)

“As a part of investigation for willingness to participate in COVID-19 vaccination research, the questionnaire of our study included three parts:1) COVID-19 awareness (perceived risk of death from COVID-19, knowledge of the transmission of COVID-19, awareness of recommended healthcare-seeking behavior); 2) sociodemographic information (sex, age, highest education level, ethnicity, residence type, health care providers, annual household income, personal history of COVID-19 diagnosis, positive acquaintance COVID-19 diagnosis); and 3) the Patient Health Questionnaire-9 (PHQ-9; a depression screening tool⁸).

We utilized the PHQ-9 depression scale as a screening tool. There are several reasons for using this tool. At only 9 items, it is suitable for a large-scale population survey and easy to be completed by the general population as a quick depression assessment. The nine diagnostic symptom criteria of the PHQ-9 correspond to the DSM-IV major depressive disorder (MDD) criteria and can facilitate follow-up review of symptoms and the diagnosis process⁹. The PHQ-9 was also selected as it has good internal consistency^{10,11}, reliability, and validity^{12,13}, with a Cronbach’s alpha coefficient between 0.80 and 0.90. We defined a score of 10 or greater as depression; COVID-19 awareness consisted of nine items which were previously published¹⁴. Among these nine items, for categorical outcomes, data are expressed as binary or categorical (range: 0 -100%). For continuous outcomes, data are expressed as median (interquartile range).”

[Comment-5] *Statistical Analysis Page 8 last sentence.*

I suggest rephrasing as “All analyses were conducted using R version 4.0.3 (R Foundation for Statistical Computing, Vienna, Austria).”

[Answer] Thank you for this comment. The authors have improved the description according to the reviewer’s suggestion. (**page 9, paragraph 1**)

“All analyses were conducted using R version 4.0.3 (R Foundation for Statistical Computing, Vienna, Austria).”

[Comment-6] *Results*

Page 9 line 17-18

Write out number 50.8 in words: Fifty-point eight percent. Do not start a sentence with Number. [NOTE FROM THE EDITORS: please feel free to rebut this comment - it is preferred to have all numbers in numerals, irrespective of the place in the sentence]

[Answer] The authors have reviewed the format of the number and used the format suggested by the editors. (**Page 9, paragraph 3**):

“The majority of participants were male (50.8%)”

[Comment-7] *Discussion*

Your sample is unlikely to be representative of the general population of China. So it is not a well-designed epidemiological study, is the term “prevalence of depression” appropriate? Depression prevalence shall be replaced by “incidence of depression”. [NOTE FROM THE EDITORS: please feel free to rebut the suggestion to replace 'prevalence' with 'incidence', as this suggestion does not seem to make sense with respect to the issue of representativeness. Please do respond to the general comment on representativeness]

[Answer] Thank you for your comment. The authors prefer to use the term “prevalence of depression” as suggested by the editors above. Prevalence is the proportion of a specific population with a particular disease during a given time, and incidence refers to the rate of new cases of a disease occurring in a specific population over a particular period of time. We did not know whether the participants with depression represent new cases, so we can only calculate the prevalence.

[Comment-8] *Why nurses are at high-risk of depression? Ps elaborate?*

[Answer] Thanks for this comment. The authors have explained the reason why nurses are at high-risk of depression and added the following text to the Discussion section. (**Page 14, paragraph 2**)

“This study also found that nurses had a higher odds of depression, consistent with previous studies¹⁵¹⁶. One possible explanation was that as front-line healthcare workers, nurses had higher risk to be infected by longer contact with patients than doctors, and worked longer hours than usual¹⁷, which might make them become more frustrated.”

[Comment-9] *This is the first study that investigated the prevalence of depression by province during the initial stage of the COVID-19 pandemic in China?*

I suggest a literature search and a literature or systematic review, or a mini-review with illustrative table including this study could add more value to the work.

[Answer] Thank you for this suggestion. The authors have improved the literature search and updated the references in the Introduction section. (**Page 5, paragraph 2**)

“In China, the estimated overall prevalence of depression was 26.9%¹⁸. In all studies on depression during the initial stage of the COVID-19 pandemic, only four studies included the general population¹⁹⁻²², and none investigated the prevalence of depression by province.”

[Comment-10] *References.*

I suggest updating the references, as several papers have now been published on depression amongst pregnant women during COVID-19.

The reference style is needed to be changed, family name first.

for example :

14

36

17 and 19 are duplicated. Please check.

[Answer] The reference section was updated as the review suggested. The authors improved reference 14 and 18, as well as deleted the reference 19. Reference 36 has been removed from the updated discussion and updated all references (**Page 19~22**).

“12. Dawel A, Shou Y, Smithson M, et al. The effect of COVID-19 on mental health and wellbeing in a representative sample of Australian adults. *Frontiers in psychiatry* 2020;11:1026.

13. Bareeqa SB, Ahmed SI, Samar SS, et al. Prevalence of depression, anxiety and stress in China during COVID-19 pandemic: A systematic review with meta-analysis. *The International Journal of Psychiatry in Medicine* 2021;56(4):210-27.

14. Al Zabadi H, Alhroub T, Yaseen N, et al. Assessment of Depression Severity During Coronavirus Disease 2019 Pandemic Among the Palestinian Population: A Growing Concern and an Immediate Consideration. *Frontiers in psychiatry* 2020;11:570065. doi: 10.3389/fpsy.2020.570065

15. Batra K, Singh TP, Sharma M, et al. Investigating the Psychological Impact of COVID-19 among Healthcare Workers: A Meta-Analysis. *Int J Environ Res Public Health* 2020;17(23) doi: 10.3390/ijerph17239096

16. Hamzeh AZ, Thair AH, Noor Y, et al. Assessment of depression severity during COVID-19 pandemic among the Palestinian population: a growing concern and an immediate consideration. *J Frontiers in psychiatry* 2020;11:1486.

17. Huang Y, Zhao N. Generalized anxiety disorder, depressive symptoms and sleep quality during COVID-19 outbreak in China: a web-based cross-sectional survey. *Psychiatry Res* 2020;288:112954. doi: 10.1016/j.psychres.2020.112954
18. Wang C, Pan R, Wan X, et al. A longitudinal study on the mental health of general population during the COVID-19 epidemic in China. *Brain, behavior, and immunity* 2020;87:40-48. doi: 10.1016/j.bbi.2020.04.028
19. Sazakli E, Leotsinidis M, Bakola M, et al. Prevalence and associated factors of anxiety and depression in students at a Greek university during COVID-19 lockdown. *Journal of public health research* 2021
20. Lei L, Huang X, Zhang S, et al. Comparison of prevalence and associated factors of anxiety and depression among people affected by versus people unaffected by quarantine during the COVID-19 epidemic in Southwestern China. *J Medical science monitor: international medical journal of experimental clinical research* 2020;26:e924609-1.
21. Durankuş F, Aksu E. Effects of the COVID-19 pandemic on anxiety and depressive symptoms in pregnant women: a preliminary study. *The Journal of Maternal-Fetal Neonatal Medicine* 2020:1-7.
22. Zhang W, Wang K, Yin L, et al. Mental health and psychosocial problems of medical health workers during the COVID-19 epidemic in China. *Psychotherapy psychosomatics* 2020;89(4):242-50.
23. Mishra P, Bhadauria US, Dasar PL, et al. Knowledge, attitude and anxiety towards pandemic flu a potential bio weapon among health professionals in Indore City. *Przegląd epidemiologiczny* 2016;70(1):41-5, 125.
24. Krok D, Zarzycka B. Risk Perception of COVID-19, Meaning-Based Resources and Psychological Well-Being amongst Healthcare Personnel: The Mediating Role of Coping. *Journal of clinical medicine* 2020;9(10):3225.
25. Floyd DL, Prentice-Dunn S, Rogers RW. A meta-analysis of research on protection motivation theory. *Journal of applied social psychology* 2000;30(2):407-29.
26. National Bureau of Statistics of China. The urban-rural composition of the population by region and the birth rate, death rate, and natural growth rate (2019) Beijing2020 [cited 2021 May 30]. Available from: <http://www.stats.gov.cn/tjsj/ndsj/2020/indexch.htm> accessed May 30 2021.
27. Kroenke K, Spitzer RL. The PHQ-9: A New Depression Diagnostic and Severity Measure. *Psychiatr Ann* 2002;32(9):509-21.
28. Fann JR, Berry DL, Wolpin S, et al. Depression screening using the Patient Health Questionnaire-9 administered on a touch screen computer. *Psycho-Oncology: Journal of the Psychological, Social Behavioral Dimensions of Cancer* 2009;18(1):14-22

Table 1. Education Level shall be converted to western style so that readers from different country can understand.

[Answer] Thank you for this comment. The education levels in Table 1 have been revised. (Page 10-12)

Table 1. Sociodemographic Characteristics, COVID-19 Awareness, and Depression Score (PHQ-9) (10,000 survey participants)

Characteristic	Not Depressed (%) (Weighted; n=9396)	Depressed (%) (Weighted; n=604)	Sample characteristics		
			Proportion (Weighted) ¹	N (Unweighted)	Proportion (Unweighted) ²
Sex					
Female	49.1%	46.2%	48.9%	4,921 (49.2%)	48.9%
Age Group					
< 20 Years	10.5%	11.0%	10.5%	900 (9.0%)	6.9%
20-29 Years	17.1%	17.9%	17.1%	1,645 (16.5%)	20.8%
30-39 Years	18.1%	15.7%	17.9%	1,895 (19.0%)	18.2%
40-49 Years	18.7%	20.5%	18.9%	1,890 (18.9%)	22.1%
50-59 Years	17.9%	20.1%	18.0%	1,820 (18.2%)	16.5%
≥ 60 Years	17.7%	14.9%	17.5%	1,675 (16.8%)	15.4%
Highest Education Level					
less than high school	24.8%	25.5%	24.8%	2792 (27.0%)	68.5%
High School / Technical Secondary School	37.1%	36.8%	37.1%	3,733 (37.3%)	17.6%
College / Undergraduate	36.0%	35.7%	36.0%	3,369 (33.7%)	13.4%
Graduate school and above	2.1%	2.0%	2.1%	206 (2.0%)	0.6%
Ethnicity					
Han	95.1%	94.8%	95.1%	9,381 (93.8%)	95.0%
Man	0.4%	0.6%	0.5%	149 (1.5%)	0.7%
Hui	0.1%	0.2%	0.1%	109 (1.1%)	0.8%

Zang	1.8%	1.5%	1.8%	103 (1.0%)	0.5%
Zhuang	1.4%	1.5%	1.4%	152 (1.5%)	1.2%
Other	1.1%	1.5%	1.1%	106 (1.1%)	1.8%
Province of Residence					
Anhui	4.6%	4.0%	4.5%	360 (3.6%)	4.5%
Beijing	1.5%	1.9%	1.5%	360 (3.6%)	1.5%
Chongqing	2.2%	2.4%	2.2%	360 (3.6%)	2.2%
Fujian	2.8%	2.9%	2.8%	300 (3.0%)	2.8%
Gansu	1.9%	1.8%	1.9%	300 (3.0%)	1.9%
Guangdong	8.1%	9.5%	8.2%	360 (3.6%)	8.2%
Guangxi	3.5%	3.8%	3.5%	300 (3.0%)	3.5%
Guizhou	2.6%	1.8%	2.6%	300 (3.0%)	2.6%
Hainan	0.7%	0.4%	0.7%	300 (3.0%)	0.7%
Hebei	5.4%	5.5%	5.4%	360 (3.6%)	5.4%
Heilongjiang	2.7%	2.1%	2.7%	300 (3.0%)	2.7%
Henan	6.6%	10.3%	6.9%	360 (3.6%)	6.9%
Hubei	4.2%	4.3%	4.2%	360 (3.6%)	4.2%
Hunan	4.9%	4.7%	4.9%	300 (3.0%)	4.9%
Jiangsu	5.7%	7.1%	5.7%	360 (3.6%)	5.7%
Jiangxi	3.4%	2.8%	3.3%	300 (3.0%)	3.3%
Jilin	1.9%	1.8%	1.9%	300 (3.0%)	1.9%
Liaoning	3.1%	2.6%	3.1%	340 (3.4%)	3.1%
Neimengol	1.9%	1.2%	1.8%	300 (3.0%)	1.8%
Ningxia	0.5%	0.4%	0.5%	300 (3.0%)	0.5%
Qinghai	0.4%	0.3%	0.4%	300 (3.0%)	0.4%
Shaanxi	2.8%	2.6%	2.7%	360 (3.6%)	2.7%
Shandong	7.3%	5.7%	7.2%	360 (3.6%)	7.2%
Shanghai	1.7%	1.4%	1.7%	300 (3.0%)	1.7%
Shanxi	2.7%	1.7%	2.8%	300 (3.0%)	2.8%
Sichuan	5.9%	7.1%	6.0%	360 (3.6%)	6.0%

Tianjin	1.1%	1.0%	1.1%	360 (3.6%)	1.1%
Tibet	0.2%	0.3%	0.3%	300 (3.0%)	0.3%
Xinjiang	1.8%	2.2%	1.8%	300 (3.0%)	1.8%
Yunnan	3.5%	2.7%	3.5%	300 (3.0%)	3.5%
Zhejiang	4.2%	3.5%	4.2%	360 (3.6%)	4.2%
Residence Type					
Urban	60.9%	62.9%	61.1%	5,935 (59.4%)	60.6%
Health Care Providers					
No	96.2%	93.6%	96.0%	9,597 (96.0%)	99.0%
Nurse	0.4%	1.3%	0.5%	55 (0.6%)	0.3%
Physician	0.8%	1.3%	0.9%	84 (0.8%)	0.5%
Community Health Worker	1.4%	2.5%	1.5%	157 (1.6%)	<0.1%
Pharmacist	0.1%	0.1%	0.1%	17 (0.2)	<0.1%
Other Healthcare Provider	0.9%	1.2%	1.0%	90 (0.9%)	0.1%
Annual Household Income (RMB)					
< 30,000	5.7%	6.5%	5.8%	560 (5.6%)	
30,000 - 59,999	15.0%	13.9%	14.9%	1,670 (16.7%)	-
60,000 - 89,999	21.7%	19.6%	21.6%	2,303 (23.0%)	-
90,000 - 119,999	25.9%	30.9%	26.2%	2,704 (24.0%)	-
120,000 - 149,999	14.2%	14.9%	14.2%	1,211 (12.1%)	-
150,000 - 199,999	11.1%	8.8%	11.1%	974 (9.7%)	-
≥ 200,000	6.4%	5.4%	6.4%	578(5.8%)	-
Personal COVID-19 Diagnosis History					
No	99.9%	99.8%	99.9%	9,992 (99.9%)	

Yes	0.1%	0.2%	0.1%	8(0.08%)
Positive Acquaintance COVID-19 Diagnosis				
Family member	0.1%	0.2%	0.1%	5(0.05%)
Friend	0.2%	0.0%	0.1%	12(0.12%)
Neighbor	0.1%	0.0%	0.1%	7(0.07%)
Cowoker	0.1%	0.3%	0.1%	10(0.1%)
Other	0.0%	0.0%	0.0%	1(0)
COVID-19 Awareness				
Perceived risk of death among vulnerable groups	3.3% (IQR: 1.0%-4.0%)	3.5% (IQR: 1.0%-4.0%)	3.3% (IQR: 1.0%-4.0%)	3.3% (IQR: 1.0%-4.0%)
Perceived risk of death among people with other diseases	96.6% (95% CI: 96.2% - 96.9%)	94.4% (95% CI: 92.2% - 96.0%)	96.4% (95% CI: 96.0% - 96.8%)	96.3% (95% CI: 96.0% - 96.7%)
Perceived elderly as a high-risk group of transmission	92.6% (95% CI: 92.0% - 93.1%)	92.6% (95% CI: 90.1% - 94.4%)	92.6% (95% CI: 92.0% - 93.1%)	92.7% (95% CI: 92.2% - 93.2%)
Awareness of vaccine availability protecting against transmission	76.7% (95% CI: 75.8% - 77.5%)	74.9% (95% CI: 71.2% - 78.1%)	76.5% (95% CI: 75.7% - 77.3%)	76.0% (95% CI: 75.1% - 76.8%)
Awareness that masks are highly effective in protecting against transmission	89.3% (95% CI: 88.7% - 89.9%)	88.0% (95% CI: 85.1% - 90.4%)	89.2% (95% CI: 88.6% - 89.8%)	89.8% (95% CI: 89.2% - 90.4%)
Knowledge of transmission prevention actions	11.6% (95% CI: 10.9% - 12.2%)	8.1% (95% CI: 6.2% - 10.6%)	11.4% (95% CI: 10.8% - 12.0%)	11.1% (95% CI: 10.5% - 11.7%)
Knowledge of saliva as main transmission route	82.6% (95% CI: 81.8% - 83.3%)	76.9% (95% CI: 73.4% - 80.1%)	82.2% (95% CI: 81.5% - 83.0%)	82.1% (95% CI: 81.3% - 82.8%)
Awareness of COVID-19 symptoms	69.4% (95% CI: 68.5% - 70.3%)	64.4% (95% CI: 60.5% - 68.1%)	69.1% (95% CI: 68.2% - 70.0%)	67.1% (95% CI: 66.1% - 68.0%)

Awareness	of	71.6%	(95%	68.2%	(95%	71.4%	68.8%
recommended healthcare-	CI:	70.7%	-	CI:	64.3%	-	CI:
seeking behavior		72.5%)			71.8%)		
					70.5%	-	67.9%
					72.3%)		69.7%)

¹ Weighted using survey sampling weights.

² As per the 2020 China Statistical Yearbook.

For dichotomous outcomes, data are expressed as a percentage with the correct response (95% confidence interval). For continuous outcomes, data are expressed as median (interquartile range).

Response to Reviewer 2

[Comment-1] *This study includes a large (10,000) sample from across each of China's provinces and participants completed a well validated depression screen. The paper is well-written, includes reference to relevant research, and analyses incorporate survey weights and examine important predictors.*

Introduction

1) *Please update COVID-19 prevalence and mortality figures (these are from June, 2021).*

[Answer] Thank you very much for this comment. The authors have updated COVID-19 prevalence and mortality figures in the introduction section. (**Page 5, paragraph 1**)

“As of middle January 2022 the World Health Organization (WHO) reports >326 million confirmed cases and >5.53 million deaths worldwide from the coronavirus disease 2019 (COVID-19) pandemic²”

[Comment-2]

2) *Update description of vaccines (being developed)*

[Answer] The authors have updated the description of COVID-19 vaccines in development. (**Page 5, paragraph 1**)

“Effective vaccines are critical to the containment of the COVID-19 pandemic; therefore, more and more vaccines are in clinical and pre-clinical development. Over 56 confirmed effective candidate vaccines for COVID-19 are being produced in America, Europe, China, and Australasia. As of middle January 2022, 139 vaccines are in clinical development, and 194 vaccines are in preclinical development preclinical phases²³. Moreover, about 59.9% of the world population has uptake at least one dose of a COVID-19 vaccine, and 9.68 billion doses have been administered worldwide, as well as 32.92 million doses are administered each day²⁴.”

[Comment-3]

3) *Description of China having returned to 'normal life' whereas other countries have not does not tally with objective indicators such as the COVID-19 Stringency Index where China is currently ranked as one of the countries with the most stringent restrictions globally (<https://ourworldindata.org/grapher/covid-stringency-index>). If China had returned fully to normal life it would also beg the question as to why it is necessary to conduct this study. Please update the introduction to reflect how China compares to other nations on indicators such as the Stringency Index.*

[Answer] Thank you for this helpful comment. The authors have followed your suggestion and have revised the description with the following sentence. (**Page 5, paragraph 1**)

“Although China is currently ranked as one of the countries with the most stringent restrictions, scoring the sixth highest in Stringency Index globally²⁵, asymptomatic infection and mutated variants continue to pose an unprecedented threat.”

[Comment-4]

4) Please include more information on how participants were recruited / why people register with KuRunData (e.g. is this an existing online internet panel for market research etc.).

[Answer] Thank you for this suggestion. The authors have expanded the explanation of how participants were recruited in the methods section and this online market research tool. (**Page 7, paragraph 1**)

“The KurunData is an online internet panel for market research, which recruits participants and provides access to the questionnaire via Wechat mini. To assure the representativeness of the data, we utilized stratified sampling by sex, residence type and province. We calculated the size of each stratum based on a sequential method (Table S1) to reflect the province’s population composition by sex and urban and rural residence (according to the China Statistical Yearbook 2020¹). When sufficient participants were recruited for a given stratum, we halted recruitment for this stratum. Participants were paid 5 yuan (US\$0.77) for completing the questionnaire. All participants completed informed written consent. Finally, 14493 participants initiated the online survey, with 10000 participants completing the survey.”

[Comment-5]

5) The typical cut-off used on the PHQ-9 is greater than or equal to 10. A cut-off of greater than or equal to 4 is unusual. Even mild depressive symptoms are typically gauged using a score of greater than or equal to 5 on this measure. Unless there is a strong rationale not to do so I suggest using the more conventional cut-off of ≥ 10 which the original Kroenke et al (2001) paper has shown to have a sensitivity of 88% and a specificity of 88% for major depression. The use of the 4 cut-off partly explains why the rate of depression appears to be very high in the current results.

[Answer] Thank you for this helpful suggestion. The authors have utilized a cut-off of ≥ 10 , as recommended by this reviewer and updated Table 1 and Table 2 (**Page 10-13**). The results and discussion sections were also updated. (**Page 9-17**)

“The majority of participants were male (50.8%), had at least a high school education (73.0%), lived in urban areas (59.4%), and had an annual household income $\geq 90,000$ RMB (~US \$13,000) (51.6%). Han ethnicity accounted for the vast majority of participants (93.8%). Except for 9% of participants aged under 20 years, the percentage of other age groups ranged from 16.5% to 19.0%. Health care providers accounted for 4.1% of participants, with the largest category being community health workers (1.6%). 0.08% of participants had been diagnosed with COVID-19 and 0.34% of participants knew someone diagnosed with COVID-19. 6.0% of participants reported being previously diagnosed with depression. The correct rate of COVID-19 awareness ranged from 11.1% to 96.3%, knowledge of COVID-19 transmission prevention actions was lowest (11.1% [95% CI: 10.5% - 11.7%]) and knowledge about perceived risk of COVID-19 death among people with other diseases was highest (96.3% [95% CI: 96.0% - 96.7%]) (See Table 1.)

Variation of depression prevalence by province

Overall, the depression prevalence was 6.3% (5.7%-6.8%). As seen in Figure 1, the prevalence of depression in Henan was the highest (9.4% [6.6%-12.7%]) and in Hainan was the lowest (3.7% [1.9%-6.1%]). The prevalence of COVID-19 in Hubei province (1,148 per million) tended to be higher than other areas in China during the same time period, and Tibet (0.28 per million) saw the lowest

prevalence (Table S2). While the prevalence of COVID-19 cases in other provinces tended to be similar, Hubei and Tibet were the exception (Figure 2).

Variation of depression prevalence by sociodemographic characteristics within provinces

The results of covariate-unadjusted and covariate-adjusted logistic regression analysis of associated factors with depression are shown in Table 2. After adjusting for covariates, urban residents and nurses had higher odds of depression than rural residents (OR, 1.50; 95% CI, 1.18 – 1.90) and other health care providers (OR, 3.06; 95% CI, 1.41-6.66). Participants who had correct knowledge of COVID-19 transmission prevention actions (OR, 0.71; 95% CI, 0.51 – 0.98) and knowledge of saliva as the main transmission route (OR, 0.80; 95% CI, 0.64 – 0.99) had lower odds of depression, as did participants who had accurate awareness of COVID-19 symptoms (OR, 0.82; 95% CI, 0.68 – 1.00).

Table 2. Sociodemographic Characteristics, COVID-19 Awareness, and Depression Score (PHQ-9) (10,000 survey participants)

Characteristic	Not Depressed (%) (Weighted; n=9396)	Depressed (%) (Weighted; n=604)	Sample characteristics		
			Proportion (Weighted) ¹	N (Unweighted)	Proportion (Unweighted) ²
Sex					
Female	49.1%	46.2%	48.9%	4,921 (49.2%)	48.9%
Age Group					
< 20 Years	10.5%	11.0%	10.5%	900 (9.0%)	6.9%
20-29 Years	17.1%	17.9%	17.1%	1,645 (16.5%)	20.8%
30-39 Years	18.1%	15.7%	17.9%	1,895 (19.0%)	18.2%
40-49 Years	18.7%	20.5%	18.9%	1,890 (18.9%)	22.1%
50-59 Years	17.9%	20.1%	18.0%	1,820 (18.2%)	16.5%
≥ 60 Years	17.7%	14.9%	17.5%	1,675 (16.8%)	15.4%
Highest Education Level					
less than high school	24.8%	25.5%	24.8%	2792 (27.0%)	68.5%

High School / Technical Secondary School	37.1%	36.8%	37.1%	3,733 (37.3%)	17.6%
College / Undergraduate	36.0%	35.7%	36.0%	3,369 (33.7%)	13.4%
Graduate school and above	2.1%	2.0%	2.1%	206 (2.0%)	0.6%
Ethnicity					
Han	95.1%	94.8%	95.1%	9,381 (93.8%)	95.0%
Man	0.4%	0.6%	0.5%	149 (1.5%)	0.7%
Hui	0.1%	0.2%	0.1%	109 (1.1%)	0.8%
Zang	1.8%	1.5%	1.8%	103 (1.0%)	0.5%
Zhuang	1.4%	1.5%	1.4%	152 (1.5%)	1.2%
Other	1.1%	1.5%	1.1%	106 (1.1%)	1.8%
Province of Residence					
Anhui	4.6%	4.0%	4.5%	360 (3.6%)	4.5%
Beijing	1.5%	1.9%	1.5%	360 (3.6%)	1.5%
Chongqing	2.2%	2.4%	2.2%	360 (3.6%)	2.2%
Fujian	2.8%	2.9%	2.8%	300 (3.0%)	2.8%
Gansu	1.9%	1.8%	1.9%	300 (3.0%)	1.9%
Guangdong	8.1%	9.5%	8.2%	360 (3.6%)	8.2%
Guangxi	3.5%	3.8%	3.5%	300 (3.0%)	3.5%
Guizhou	2.6%	1.8%	2.6%	300 (3.0%)	2.6%
Hainan	0.7%	0.4%	0.7%	300 (3.0%)	0.7%
Hebei	5.4%	5.5%	5.4%	360 (3.6%)	5.4%
Heilongjiang	2.7%	2.1%	2.7%	300 (3.0%)	2.7%
Henan	6.6%	10.3%	6.9%	360 (3.6%)	6.9%
Hubei	4.2%	4.3%	4.2%	360 (3.6%)	4.2%
Hunan	4.9%	4.7%	4.9%	300 (3.0%)	4.9%
Jiangsu	5.7%	7.1%	5.7%	360 (3.6%)	5.7%
Jiangxi	3.4%	2.8%	3.3%	300 (3.0%)	3.3%

Jilin	1.9%	1.8%	1.9%	300 (3.0%)	1.9%
Liaoning	3.1%	2.6%	3.1%	340 (3.4%)	3.1%
Neimengol	1.9%	1.2%	1.8%	300 (3.0%)	1.8%
Ningxia	0.5%	0.4%	0.5%	300 (3.0%)	0.5%
Qinghai	0.4%	0.3%	0.4%	300 (3.0%)	0.4%
Shaanxi	2.8%	2.6%	2.7%	360 (3.6%)	2.7%
Shandong	7.3%	5.7%	7.2%	360 (3.6%)	7.2%
Shanghai	1.7%	1.4%	1.7%	300 (3.0%)	1.7%
Shanxi	2.7%	1.7%	2.8%	300 (3.0%)	2.8%
Sichuan	5.9%	7.1%	6.0%	360 (3.6%)	6.0%
Tianjin	1.1%	1.0%	1.1%	360 (3.6%)	1.1%
Tibet	0.2%	0.3%	0.3%	300 (3.0%)	0.3%
Xinjiang	1.8%	2.2%	1.8%	300 (3.0%)	1.8%
Yunnan	3.5%	2.7%	3.5%	300 (3.0%)	3.5%
Zhejiang	4.2%	3.5%	4.2%	360 (3.6%)	4.2%
Residence Type					
Urban	60.9%	62.9%	61.1%	5,935 (59.4%)	60.6%
Health Care Providers					
No	96.2%	93.6%	96.0%	9,597 (96.0%)	99.0%
Nurse	0.4%	1.3%	0.5%	55 (0.6%)	0.3%
Physician	0.8%	1.3%	0.9%	84 (0.8%)	0.5%
Community Health Worker	1.4%	2.5%	1.5%	157 (1.6%)	<0.1%
Pharmacist	0.1%	0.1%	0.1%	17 (0.2)	<0.1%
Other Healthcare Provider	0.9%	1.2%	1.0%	90 (0.9%)	0.1%
Annual Household Income (RMB)					
< 30,000	5.7%	6.5%	5.8%	560 (5.6%)	
30,000 - 59,999	15.0%	13.9%	14.9%	1,670 (16.7%)	-

60,000 - 89,999	21.7%	19.6%	21.6%	2,303 (23.0%)	-
90,000 - 119,999	25.9%	30.9%	26.2%	2,704 (24.0%)	-
120,000 - 149,999	14.2%	14.9%	14.2%	1,211 (12.1%)	-
150,000 - 199,999	11.1%	8.8%	11.1%	974 (9.7%)	-
≥ 200,000	6.4%	5.4%	6.4%	578(5.8%)	-
Personal COVID-19 Diagnosis History					
No	99.9%	99.8%	99.9%	9,992 (99.9%)	
Yes	0.1%	0.2%	0.1%	8(0.08%)	
Postive Acquaintance COVID-19 Diagnosis					
Family member	0.1%	0.2%	0.1%	5(0.05%)	
Friend	0.2%	0.0%	0.1%	12(0.12%)	
Neighbor	0.1%	0.0%	0.1%	7(0.07%)	
Cowoker	0.1%	0.3%	0.1%	10(0.1%)	
Other	0.0%	0.0%	0.0%	1(0)	
COVID-19 Awareness					
Perceived risk of death among vulnerable groups	3.3% (IQR: 1.0%-4.0%)	3.5% (IQR: 1.0%-4.0%)	3.3% (IQR: 1.0%-4.0%)	3.3% (IQR: 1.0%-4.0%)	
Perceived risk of death among people with other diseases	96.6% (95% CI: 96.2% - 96.9%)	94.4% (95% CI: 92.2% - 96.0%)	96.4% (95% CI: 96.0% - 96.8%)	96.3% (95% CI: 96.0% - 96.7%)	
Perceived elderly as a high-risk group of transmission	92.6% (95% CI: 92.0% - 93.1%)	92.6% (95% CI: 90.1% - 94.4%)	92.6% (95% CI: 92.0% - 93.1%)	92.7% (95% CI: 92.2% - 93.2%)	
Awareness of vaccine availability protecting against transmission	76.7% (95% CI: 75.8% - 77.5%)	74.9% (95% CI: 71.2% - 78.1%)	76.5% (95% CI: 75.7% - 77.3%)	76.0% (95% CI: 75.1% - 76.8%)	

Awareness that masks are highly effective in protecting against transmission	89.3% (95% CI: 88.7% - 89.9%)	88.0% (95% CI: 85.1% - 90.4%)	89.2% (95% CI: 88.6% - 89.8%)	89.8% (95% CI: 89.2% - 90.4%)
Knowledge of transmission prevention actions	11.6% (95% CI: 10.9% - 12.2%)	8.1% (95% CI: 6.2% - 10.6%)	11.4% (95% CI: 10.8% - 12.0%)	11.1% (95% CI: 10.5% - 11.7%)
Knowledge of saliva as main transmission route	82.6% (95% CI: 81.8% - 83.3%)	76.9% (95% CI: 73.4% - 80.1%)	82.2% (95% CI: 81.5% - 83.0%)	82.1% (95% CI: 81.3% - 82.8%)
Awareness of COVID-19 symptoms	69.4% (95% CI: 68.5% - 70.3%)	64.4% (95% CI: 60.5% - 68.1%)	69.1% (95% CI: 68.2% - 70.0%)	67.1% (95% CI: 66.1% - 68.0%)
Awareness of recommended healthcare-seeking behavior	71.6% (95% CI: 70.7% - 72.5%)	68.2% (95% CI: 64.3% - 71.8%)	71.4% (95% CI: 70.5% - 72.3%)	68.8% (95% CI: 67.9% - 69.7%)

¹ Weighted using survey sampling weights.

² As per the 2020 China Statistical Yearbook.

For dichotomous outcomes, data are expressed as a percentage with the correct response (95% confidence interval). For continuous outcomes, data are expressed as median (interquartile range).

Table 3. Covariate-unadjusted and covariate-adjusted logistic regressions of depression onto sociodemographic characteristics

	Covariate-unadjusted OR (95% CI) ¹	Covariate-adjusted OR (95% CI) ²	Depression prevalence (95% CI)
Overall			604 (6.3%)[5.7%~6.8%]
Sex			
Male	Ref.	Ref.	323 (6.6%)[5.8%~7.4%]

Female	0.90 (0.75 - 1.07)	0.89 (0.74 - 1.07)	281 (5.9%)[5.2%~6.7%]
Age			
18-19 years	Ref.	Ref.	65 (6.5%)[4.9%~8.4%]
20-29 years	1.01 (0.72 - 1.42)	1.06 (0.72 - 1.56)	117 (6.5%)[5.2%~8.0%]
30-39 years	0.85 (0.59 - 1.21)	0.91 (0.60 - 1.37)	98 (5.5%)[4.3%~6.8%]
40-49 years	1.07 (0.76 - 1.49)	1.12 (0.77 - 1.63)	114 (6.8%)[5.5%~8.3%]
50-59 years	1.09 (0.78 - 1.53)	1.05 (0.72 - 1.54)	120 (7.0%)[5.7%~8.4%]
>60 years	0.83 (0.58 - 1.19)	0.82 (0.52 - 1.31)	90 (5.3%)[4.2%~6.6%]
Highest Education Level			
less than high school	Ref	Ref	165 (7.2%)[5.8%~8.7%]
High school / Technical secondary school	1.48 (0.85 - 2.59)	1.16 (0.61 - 2.21)	225 (6.2%)[5.3%~7.2%]
College / Undergraduate	1.45 (0.83 - 2.54)	1.12 (0.58 - 2.14)	202 (6.2%)[5.3%~7.2%]
Graduate school and above	1.37 (0.58 - 3.25)	1.09 (0.43 - 2.75)	12 (5.9%)[2.6%~10.5%]
Residence Type			
Rural	Ref.	Ref.	243 (6.0%)[5.1%~6.9%]
Urban	1.09 (0.90 - 1.31)	1.50 (1.18 - 1.90)**	361 (6.5%)[5.8%~7.2%]
Health Care Providers			
No	Ref.	Ref.	564 (6.1%)[5.6%~6.7%]
Nurse	2.61 (1.18 - 5.75)*	3.06 (1.41 - 6.66)**	7 (16.1%)[5.1%~31.5%]
Physician	1.48 (0.65 - 3.38)	1.51 (0.67 - 3.40)	7 (9.2%)[2.9%~18.5%]

Community health worker	1.66 (0.97 - 2.86)	1.62 (0.94 - 2.78)	16 (10.4%)[5.5%~16.6%]
Pharmacist	0.91 (0.18 - 4.71)	1.17 (0.24 - 5.69)	2 (11.8%)[NA~NA]
Other healthcare provider	1.38 (0.65 - 2.92)	1.31 (0.60 - 2.90)	8 (8.1%)[2.9%~15.5%]
Annual household income (RMB)			
< 30,000	Ref.	Ref.	39 (7.1%)[4.8%~9.8%]
30,000 – 59,999	0.87 (0.58 - 1.31)	0.92 (0.60 - 1.42)	98 (5.8%)[4.6%~7.2%]
60,000 – 89,999	0.82 (0.55 - 1.22)	0.87 (0.56 - 1.34)	132 (5.7%)[4.7%~6.9%]
90,000 – 119,999	1.05 (0.71 - 1.56)	1.14 (0.74 - 1.75)	182 (7.4%)[6.3%~8.7%]
120,000 – 149,999	0.87 (0.56 - 1.34)	0.92 (0.56 - 1.49)	78 (6.6%)[5.1%~8.2%]
150,000 – 199,999	0.68 (0.42 - 1.10)	0.70 (0.41 - 1.20)	44 (5.0%)[3.6%~6.7%]
≥ 200,000	0.69 (0.40 - 1.20)	0.72 (0.39 - 1.32)	31 (5.3%)[3.4%~7.7%]
COVID-19 Awareness			
Perceived risk of death among vulnerable groups	1.02 (0.99 - 1.05)	1.02 (0.99 - 1.05)	NA
Perceived risk of death among people with other diseases	0.87 (0.72 - 1.05)	0.88 (0.73 - 1.07)	395 (6.0%)[5.4%~6.7%]
Perceived elderly as a high-risk group of transmission	1.00 (0.71 - 1.40)	1.05 (0.75 - 1.47)	558 (6.3%)[5.7%~6.9%]
Awareness of vaccine availability protecting against transmission	0.92 (0.75 - 1.13)	0.91 (0.73 - 1.13)	447 (6.1%)[5.5%~6.8%]
Awareness that masks are highly effective in protecting against transmission	0.88 (0.67 - 1.16)	0.89 (0.68 - 1.17)	533 (6.2%)[5.6%~6.8%]
Knowledge of transmission prevention actions	0.68 (0.49 - 0.94)*	0.71 (0.51 - 0.98)*	50 (4.5%)[3.2%~6.0%]
Knowledge of saliva as main transmission route	0.74 (0.60 - 0.91)**	0.80 (0.64 - 0.99)*	464 (5.9%)[5.3%~6.5%]

Awareness of COVID-19 symptoms	0.81 (0.67 - 0.97)*	0.82 (0.68 - 1.00)*	386 (5.8%)[5.2%~6.5%]
Awareness of recommended healthcare - seeking behavior	0.84 (0.69 - 1.02)	0.81 (0.65 - 1.01)	404 (6.0%)[5.4%~6.7%]
Postive Acquaintance COVID-19 Diagnosis			
No	Ref.	Ref.	602 (6.3%)[5.7%~6.8%]
Yes	1.60 (0.44 - 5.84)	1.22 (0.32 - 4.61)	2 (10.0%)[0.5%~29.5%]
COVID-19 cases confirmed by the province			
Prevalence of COVID-19 by province	1.00 (0.99 - 1.01)	1.00 (0.99 - 1.02)	NA

1. Covariate-unadjusted logistic Regression results [All regressions included each one of the variables shown in the table and adjusted for each province (province-level fixed effects)]
2. Covariate-adjusted logistic Regression results [All regressions included all variables shown in the table and adjusted for each province (province-level fixed effects)]
3. ***represents $p < 0.001$; **represents $p < 0.01$; * represents $p < 0.05$

Discussion

In our online survey sample, 6.3% of adults had depression as defined by a PHQ-9 ≥ 10 . We found a higher prevalence of depression among certain population groups, including urban residents and nurses. Furthermore, we found that knowledge and awareness of COVID-19 were associated with lower odds of depression. Under the strong assumption that these associations might be causal, this finding could indicate a “protective effect” for mental health and thus indicate the importance of effective communication and education about COVID-19 amid the pandemic.

The correlation between urban residence and depression was positive in our study, which is consistent with several previous studies^{26 27}. A possible interpretation of this finding is that while the virus could be transmitted more quickly in urban areas with a higher density population²⁸, those in urban areas tend to have higher education levels and greater access to the latest report on the COVID-19 pandemic²⁶. Another explanation is that depression was more common in the urban than in rural areas in China before the COVID-19 pandemic. Moreover, the social distancing restrictions allowed less travel in cities than in rural areas, potentially contributing to the higher prevalence of depression in urban than in rural areas. This study also found that nurses had higher odds of depression, consistent with previous studies^{15 16}. One possible explanation was that as front-line healthcare workers, nurses had higher risk to be infected by longer contact with patients than doctors, and worked longer hours than usual¹⁷, which might make them become more frustrated.

Several studies have described the prevalence of depression during the COVID-19 pandemic in the general population. Among studies using the PHQ-9 scale with the same cut-off value (≥ 10), the prevalence of depression observed in our study (6.3%) was lower than a national study among 56,679 participants conducted February 28 to March 11, 2020 in China (10.8%)²⁹ as well as a study of 1,470 individuals among the general population during the COVID-19 outbreak in the United States from March 31 to April 13, 2020 (27.8%)³⁰. Most of the other studies were conducted in February 2020—at the peak of the COVID-19 epidemic, which ranged from 8.3 to 48.3%^{29 31-33}. One possible explanation is that our study occurred in May when the epidemic had decreased in severity, and the prevalence of depression may have diminished. Another possible reason for a lower depression prevalence found in our study may be that with the continuous strict quarantine policy, the government provided timely mental health service in response to COVID-19³⁴.

The prevalence of COVID-19 in 31 provinces tended to be similar except for Hubei, which had the highest prevalence (Figure 2). However, the proportion of the population by province with depression (PHQ-9 ≥ 10) did not seem to be significantly associated with the prevalence of COVID-19 cases. Moreover, from the covariate-adjusted logistic regression results, we found that depression was not associated with the prevalence of COVID-19 cases confirmed by province, which also suggested that

the prevalence of depression is not associated with the severity of the regional epidemic during the COVID-19 outbreak. One explanation is that resilience plays a protective role in mitigating the impact of stress and trauma on depressive symptoms and the consequences associated with depressive symptoms³⁵. During the initial phase of the COVID-19 outbreak in China, the Chinese government enforced a rigid social distancing policy through social media and strongly promoted hand washing, surface disinfection, and the use of protective masks³⁶. In Wuhan, as the most affected city, people with mild and asymptomatic infection received care in Fangcang shelter hospitals, for facility-based isolation, treatment, and monitoring ³⁶, an effective method to control the epidemic ^{37 38}. Fangcang shelter hospitals also provided mental health counseling services and social support to help patients recover during isolation ³⁹, which may also reduce the incidence of depression in hard-hit areas. Due to very limited confirmed cases in our study, it was difficult to investigate the association between depression and the number of confirmed cases (0.1% participants diagnosed with COVID-19, 0.48% participants with an acquaintance with confirmed COVID-19).

This study, for the first time, provides preliminary evidence that knowledge of COVID-19 transmission prevention actions, knowledge of saliva as main transmission route and awareness of COVID-19 symptoms may be associated with a lower risk of depression. Compared to previous research^{22 26 40}, Our study reinforced the positive association between precautionary measures as well as awareness of COVID-19 symptoms and depression. One possible explanation could be that accurate knowledge of COVID-19 transmission prevention actions and awareness of COVID-19 symptoms may help alleviate the stress caused by fear of COVID-19 pandemic. Stress, as a risk factor for depression, plays a role in triggering depression by biological mechanisms such as HPA axis stress response processes and hormonal and neurotransmitter systems ⁴¹. Therefore, accurate knowledge and perceptions towards COVID-19 could indirectly affect depression through the reduction in stress. This finding suggests that, to reduce the prevalence of depression, effective communication and education of COVID-19 preventive measures and recommended healthcare-seeking behaviors are urgently needed.”

[Comment-6]

6) Please provide more information on how the sampling weights were calculated. The method section indicates that "The sampling weights were calculated by the inverse of the probability of age, residence type, and sex of selecting participants." Yet, looking at Table 1 the weights appear to produce an allignment with the China 2020 Statistical Yearbook based on Sex and Province of Residence. What role did age and residence type play in the weighting model?

[Answer] Thank you for your question. The authors did not use age for calculating the weighted model. The text has been improved to describe the sampling weight in the methods section in further detail. (Page 7, paragraph 1 and Page 8, paragraph 2)

"We calculated the size of each stratum based on a sequential method (Table S1) to reflect the province's population composition by gender and urban and rural residence (according to the China Statistical Yearbook 2020 ¹) and assure the sufficient samples collected in each stratum. Once we have recruited enough samples for a stratum, we would stop recruiting samples for this stratum.

"The sampling weights were calculated from the 2019 population census and the sampling quotas, accounting for some feature of the survey, including oversampling for sex, residence type and province. Specifically, sampling weights are the inverse of the probability of selecting participants with some specific residence type (urban or rural), sex (male or female), in some specific province among the population. We used sampling weights adjusted for the survey design to calculate statistics. For binary and categorical response options of knowledge about COVID-19, we calculated the percentage of participants with correct responses."

[Comment-7]

A table such as Table 1 including the weighted prevalence of depression by sociodemographic characteristics would be worth including.

[Answer] Thank you for this suggestion. The authors added the weighted prevalence of depression by sociodemographic characteristics in Table 1. (Page 10-12)

"Table 4. Sociodemographic Characteristics, COVID-19 Awareness, and Depression Score (PHQ-9) (10,000 survey participants)

Characteristic	Not Depressed (%) (Weighted; n=9396)	Depressed (%) (Weighted; n=604)	p value for difference	Total		
				Proportion (Weighted) ¹	N (%) (Not weighted)	Proportion ²
Sex						
Female	49.1%	46.2%	-	48.9%	4,921 (49.2%)	48.9%
Age Group						
< 20 Years	10.5%	11.0%	-	10.5%	900 (9.0%)	6.9%
20-29 Years	17.1%	17.9%	-	17.1%	1,645 (16.5%)	20.8%
30-39 Years	18.1%	15.7%	-	17.9%	1,895 (19.0%)	18.2%

40-49 Years	18.7%	20.5%	-	18.9%	1,890 (18.9%)	22. 1%
50-59 Years	17.9%	20.1%	-	18.0%	1,820 (18.2%)	16. 5%
≥ 60 Years	17.7%	14.9%	-	17.5%	1,675 (16.8%)	15. 4%
Highest Education Level						
less than high school	24.8%	25.5%	0.034	24.8%	2792 (27.0%)	68. 5%
High School / Technical Secondary School	37.1%	36.8%	-	37.1%	3,733 (37.3%)	17. 6%
College / Undergraduate	36.0%	35.7%	-	36.0%	3,369 (33.7%)	13. 4%
Graduate school and above	2.1%	2.0%	-	2.1%	206 (2.0%)	0.6 %
Ethnicity						
Han	95.1%	94.8%	-	95.1%	9,381 (93.8%)	95. 0%
Man	0.4%	0.6%	-	0.5%	149 (1.5%)	0.7 %
Hui	0.1%	0.2%	-	0.1%	109 (1.1%)	0.8 %
Zang	1.8%	1.5%	-	1.8%	103 (1.0%)	0.5 %
Zhuang	1.4%	1.5%	-	1.4%	152 (1.5%)	1.2 %
Other	1.1%	1.5%	-	1.1%	106 (1.1%)	1.8 %
Province of Residence						
Anhui	4.6%	4.0%	-	4.5%	360 (3.6%)	4.5 %
Beijing	1.5%	1.9%	-	1.5%	360 (3.6%)	1.5 %
Chongqing	2.2%	2.4%	-	2.2%	360 (3.6%)	2.2 %

Fujian	2.8%	2.9%	-	2.8%	300 (3.0%)	2.8 %
Gansu	1.9%	1.8%	-	1.9%	300 (3.0%)	1.9 %
Guangdong	8.1%	9.5%	-	8.2%	360 (3.6%)	8.2 %
Guangxi	3.5%	3.8%	-	3.5%	300 (3.0%)	3.5 %
Guizhou	2.6%	1.8%	-	2.6%	300 (3.0%)	2.6 %
Hainan	0.7%	0.4%	-	0.7%	300 (3.0%)	0.7 %
Hebei	5.4%	5.5%	-	5.4%	360 (3.6%)	5.4 %
Heilongjiang	2.7%	2.1%	-	2.7%	300 (3.0%)	2.7 %
Henan	6.6%	10.3%	0.004	6.9%	360 (3.6%)	6.9 %
Hubei	4.2%	4.3%	-	4.2%	360 (3.6%)	4.2 %
Hunan	4.9%	4.7%	-	4.9%	300 (3.0%)	4.9 %
Jiangsu	5.7%	7.1%	-	5.7%	360 (3.6%)	5.7 %
Jiangxi	3.4%	2.8%	-	3.3%	300 (3.0%)	3.3 %
Jilin	1.9%	1.8%	-	1.9%	300 (3.0%)	1.9 %
Liaoning	3.1%	2.6%	-	3.1%	340 (3.4%)	3.1 %
Neimengol	1.9%	1.2%	-	1.8%	300 (3.0%)	1.8 %
Ningxia	0.5%	0.4%	-	0.5%	300 (3.0%)	0.5 %
Qinghai	0.4%	0.3%	-	0.4%	300 (3.0%)	0.4 %
Shaanxi	2.8%	2.6%	-	2.7%	360 (3.6%)	2.7 %

Shandong	7.3%	5.7%	-	7.2%	360 (3.6%)	7.2 %
Shanghai	1.7%	1.4%	-	1.7%	300 (3.0%)	1.7 %
Shanxi	2.7%	1.7%	-	2.8%	300 (3.0%)	2.8 %
Sichuan	5.9%	7.1%	-	6.0%	360 (3.6%)	6.0 %
Tianjin	1.1%	1.0%	-	1.1%	360 (3.6%)	1.1 %
Tibet	0.2%	0.3%	-	0.3%	300 (3.0%)	0.3 %
Xinjiang	1.8%	2.2%	-	1.8%	300 (3.0%)	1.8 %
Yunnan	3.5%	2.7%	-	3.5%	300 (3.0%)	3.5 %
Zhejiang	4.2%	3.5%	-	4.2%	360 (3.6%)	4.2 %
Residence Type						
Urban	60.9%	62.9%	-	61.1%	5,935 (59.4%)	60. 6%
Health Care Providers						
No	96.2%	93.6%	0.011	96.0%	9,597 (96.0%)	99. 0%
Nurse	0.4%	1.3%	-	0.5%	55 (0.6%)	0.3 %
Physician	0.8%	1.3%	-	0.9%	84 (0.8%)	0.5 %
Community Worker	1.4%	2.5%	-	1.5%	157 (1.6%)	<0. 1%
Pharmacist	0.1%	0.1%	-	0.1%	17 (0.2)	<0. 1%
Other Provider	0.9%	1.2%	-	1.0%	90 (0.9%)	0.1 %
Annual Household Income (RMB)						

< 30,000	5.7%	6.5%	-	5.8%	560 (5.6%)	
30,000 - 59,999	15.0%	13.9%	-	14.9%	1,670 (16.7%)	-
60,000 - 89,999	21.7%	19.6%	-	21.6%	2,303 (23.0%)	-
90,000 - 119,999	25.9%	30.9%	-	26.2%	2,704 (24.0%)	-
120,000 - 149,999	14.2%	14.9%	-	14.2%	1,211 (12.1%)	-
150,000 - 199,999	11.1%	8.8%	-	11.1%	974 (9.7%)	-
≥ 200,000	6.4%	5.4%	-	6.4%	578(5.8%)	-
Personal COVID-19 Diagnosis History						
No	99.9%	99.8%	-	99.9%	9,992 (99.9%)	
Yes	0.1%	0.2%	-	0.1%	8(0.08%)	
Postive Acquaintance COVID-19 Diagnosis						
Family member	0.1%	0.2%	-	0.1%	5(0.05%)	
Friend	0.2%	0.0%	-	0.1%	12(0.12%)	
Neighbor	0.1%	0.0%	-	0.1%	7(0.07%)	
Cowoker	0.1%	0.3%	-	0.1%	10(0.1%)	
Other	0.0%	0.0%	-	0.0%	1(0)	
COVID-19 Awareness						
Perceived risk of death among vulnerable groups	3.3% (IQR: 1.0%-4.0%)	3.5% (IQR: 1.0%-4.0%)	NA	3.3% (IQR: 1.0%-4.0%)	3.3% (IQR: 1.0%-4.0%)	
Perceived risk of death among people with other diseases	96.6% (95% CI: 96.2% - 96.9%)	94.4% (95% CI: 92.2% - 96.0%)	-	96.4% (95% CI: 96.0% - 96.8%)	96.3% (95% CI: 96.0% - 96.7%)	
Perceived elderly as a high-risk group of transmission	92.6% (95% CI: 92.0% - 93.1%)	92.6% (95% CI: -	-	92.6% (95% CI: -	92.7% (95% CI: -	

		90.1%	-		92.0%	-	92.2%	-
		94.4%)			93.1%)		93.2%)	
Awareness of vaccine availability protecting against transmission	76.7% (95% CI: 75.8% - 77.5%)	74.9% (95% CI: 71.2% - 78.1%)	CI: -		76.5% (95% CI: 75.7% - 77.3%)	CI: -	76.0% (95% CI: 75.1% - 76.8%)	
Awareness that masks are highly effective in protecting against transmission	89.3% (95% CI: 88.7% - 89.9%)	88.0% (95% CI: 85.1% - 90.4%)	CI: -		89.2% (95% CI: 88.6% - 89.8%)	CI: -	89.8% (95% CI: 89.2% - 90.4%)	
Knowledge of transmission prevention actions	11.6% (95% CI: 10.9% - 12.2%)	8.1% (95% CI: 6.2% - 10.6%)	0.003		11.4% (95% CI: 10.8% - 12.0%)	CI: -	11.1% (95% CI: 10.5% - 11.7%)	
Knowledge of saliva as main transmission route	82.6% (95% CI: 81.8% - 83.3%)	76.9% (95% CI: 73.4% - 80.1%)	0.001		82.2% (95% CI: 81.5% - 83.0%)	CI: -	82.1% (95% CI: 81.3% - 82.8%)	
Awareness of COVID-19 symptoms	69.4% (95% CI: 68.5% - 70.3%)	64.4% (95% CI: 60.5% - 68.1%)	0.013		69.1% (95% CI: 68.2% - 70.0%)	CI: -	67.1% (95% CI: 66.1% - 68.0%)	
Awareness of recommended healthcare-seeking behavior	71.6% (95% CI: 70.7% - 72.5%)	68.2% (95% CI: 64.3% - 71.8%)	CI: -		71.4% (95% CI: 70.5% - 72.3%)	CI: -	68.8% (95% CI: 67.9% - 69.7%)	

¹ Weighted using survey sampling weights.

² As per the 2020 China Statistical Yearbook.

For dichotomous outcomes, data are expressed as a percentage with the correct response (95% confidence interval). For continuous outcomes, data are expressed as median (interquartile range).

[Comment-8]

8) It is not clear how you are finding such large odds ratios for family and neighbor confirmed COVID ("and having a family member (OR, 23.75; 95% CI, 1.86-303.51) or neighbor (OR, 19.71; 95% CI, 1.54 - 252.22)"). Can you give details of the prevalence of depression in these groups and demonstrate how the ORs are accurate? I would suggest combining these groups as they are very low N which may be producing these extreme ORs (e.g. Any Positive Acquaintance COVID-19 Diagnosis would be 35 of 10,000 participants or 0.35%).

[Answer] Thanks for this comment. The authors updated all results using a score of 10 or greater as depression symptoms definition in Table 2 and gave details of the prevalence of depression in these

groups for positive acquaintance COVID-19 diagnosis. The cases of neighbor and friend groups were close to 0, which may be producing extreme ORs, we combined these two groups in Table 2 as suggested. (Page 12-13)

“Table 5. Covariate-unadjusted and covariate-adjusted logistic regressions of depression onto sociodemographic characteristics

	Covariate-unadjusted OR (95% CI)¹	Covariate-adjusted OR (95% CI)²	Depression prevalence (95% CI)
Overall			604 (6.3%)[5.7%~6.8%]
Sex			
Male	Ref.	Ref.	323 (6.6%)[5.8%~7.4%]
Female	0.90 (0.75 - 1.07)	0.89 (0.74 - 1.07)	281 (5.9%)[5.2%~6.7%]
Age			
18-19 years	Ref.	Ref.	65 (6.5%)[4.9%~8.4%]
20-29 years	1.01 (0.72 - 1.42)	1.06 (0.72 - 1.56)	117 (6.5%)[5.2%~8.0%]
30-39 years	0.85 (0.59 - 1.21)	0.91 (0.60 - 1.37)	98 (5.5%)[4.3%~6.8%]
40-49 years	1.07 (0.76 - 1.49)	1.12 (0.77 - 1.63)	114 (6.8%)[5.5%~8.3%]
50-59 years	1.09 (0.78 - 1.53)	1.05 (0.72 - 1.54)	120 (7.0%)[5.7%~8.4%]
>60 years	0.83 (0.58 - 1.19)	0.82 (0.52 - 1.31)	90 (5.3%)[4.2%~6.6%]
Highest Education Level			
less than high school	Ref	Ref	165 (7.2%)[5.8%~8.7%]
High school / Technical secondary school	1.48 (0.85 - 2.59)	1.16 (0.61 - 2.21)	225 (6.2%)[5.3%~7.2%]
College / Undergraduate	1.45 (0.83 - 2.54)	1.12 (0.58 - 2.14)	202 (6.2%)[5.3%~7.2%]
Graduate school and above	1.37 (0.58 - 3.25)	1.09 (0.43 - 2.75)	12 (5.9%)[2.6%~10.5%]

Residence Type

Rural	Ref.	Ref.	243 (6.0%)[5.1%~6.9%]
Urban	1.09 (0.90 - 1.31)	1.50 (1.18 - 1.90)**	361 (6.5%)[5.8%~7.2%]

Health Care Providers

No	Ref.	Ref.	564 (6.1%)[5.6%~6.7%]
Nurse	2.61 (1.18 - 5.75)*	3.06 (1.41 - 6.66)**	7 (16.1%)[5.1%~31.5%]
Physician	1.48 (0.65 - 3.38)	1.51 (0.67 - 3.40)	7 (9.2%)[2.9%~18.5%]
Community health worker	1.66 (0.97 - 2.86)	1.62 (0.94 - 2.78)	16 (10.4%)[5.5%~16.6%]
Pharmacist	0.91 (0.18 - 4.71)	1.17 (0.24 - 5.69)	2 (11.8%)[NA~NA]
Other healthcare provider	1.38 (0.65 - 2.92)	1.31 (0.60 - 2.90)	8 (8.1%)[2.9%~15.5%]

Annual household income (RMB)

< 30,000	Ref.	Ref.	39 (7.1%)[4.8%~9.8%]
30,000 – 59,999	0.87 (0.58 - 1.31)	0.92 (0.60 - 1.42)	98 (5.8%)[4.6%~7.2%]
60,000 – 89,999	0.82 (0.55 - 1.22)	0.87 (0.56 - 1.34)	132 (5.7%)[4.7%~6.9%]
90,000 – 119,999	1.05 (0.71 - 1.56)	1.14 (0.74 - 1.75)	182 (7.4%)[6.3%~8.7%]
120,000 – 149,999	0.87 (0.56 - 1.34)	0.92 (0.56 - 1.49)	78 (6.6%)[5.1%~8.2%]
150,000 – 199,999	0.68 (0.42 - 1.10)	0.70 (0.41 - 1.20)	44 (5.0%)[3.6%~6.7%]
≥ 200,000	0.69 (0.40 - 1.20)	0.72 (0.39 - 1.32)	31 (5.3%)[3.4%~7.7%]

COVID-19 Awareness

Perceived risk of death among vulnerable groups	1.02 (0.99 - 1.05)	1.02 (0.99 - 1.05)	NA
--	-----------------------	-----------------------	----

Perceived risk of death among people with other diseases	0.87 (0.72 - 1.05)	0.88 (0.73 - 1.07)	395 (6.0%)[5.4%~6.7%]
Perceived elderly as a high-risk group of transmission	1.00 (0.71 - 1.40)	1.05 (0.75 - 1.47)	558 (6.3%)[5.7%~6.9%]
Awareness of vaccine availability protecting against transmission	0.92 (0.75 - 1.13)	0.91 (0.73 - 1.13)	447 (6.1%)[5.5%~6.8%]
Awareness that masks are highly effective in protecting against transmission	0.88 (0.67 - 1.16)	0.89 (0.68 - 1.17)	533 (6.2%)[5.6%~6.8%]
Knowledge of transmission prevention actions	0.68 (0.49 - 0.94)*	0.71 (0.51 - 0.98)*	50 (4.5%)[3.2%~6.0%]
Knowledge of saliva as main transmission route	0.74 (0.60 - 0.91)**	0.80 (0.64 - 0.99)*	464 (5.9%)[5.3%~6.5%]
Awareness of COVID-19 symptoms	0.81 (0.67 - 0.97)*	0.82 (0.68 - 1.00)*	386 (5.8%)[5.2%~6.5%]
Awareness of recommended healthcare - seeking behavior	0.84 (0.69 - 1.02)	0.81 (0.65 - 1.01)	404 (6.0%)[5.4%~6.7%]

Postive Acquaintance COVID-19 Diagnosis

No	Ref.	Ref.	602 (6.3%)[5.7%~6.8%]
Yes	1.60 (0.44 - 5.84)	1.22 (0.32 - 4.61)	2 (10.0%)[0.5%~29.5%]

COVID-19 cases confirmed by the province

Prevalence of COVID-19 by province	1.00 (0.99 - 1.01)	1.00 (0.99 - 1.02)	NA
---	--------------------	--------------------	----

1. Covariate-unadjusted logistic Regression results [All regressions included each one of the variables shown in the table and adjusted for each province (province-level fixed effects)]

2. Covariate-adjusted logistic Regression results [All regressions included all variables shown in the table and adjusted for each province (province-level fixed effects)]

3. ***represents $p < 0.001$; **represents $p < 0.01$; * represents $p < 0.05$

[Comment-9]

9) The discussion begins "This is the first study that quantified the prevalence of depression symptoms and its variation by province and sociodemographic characteristics in China". Where in the paper was the prevalence of depression quantified by province?

[Answer] The description of the prevalence of depression quantified by province appears in the results section. (Page 10, paragraph 1) and Table 1(Page 10-12).

“Variation of depression prevalence by province

As seen in Figure 1, the prevalence of depression (PHQ-9 score ≥10) in southern China tended to be greater than in northern China. The overall prevalence of depression was 6.0% (95% CI: 5.6%-6.5%). The prevalence of COVID-19 in Hubei province (1,148 per million) tended to be higher than other areas in China, and Tibet (0.28 per million) was the lowest province. While the prevalence of COVID-19 cases from other provinces tended to be the same except for Hubei and Tibet (Figure 2).”

“Table 6. Sociodemographic Characteristics, COVID-19 Awareness, and Depression Score (PHQ-9) (10,000 survey participants)

Characteristic	Not Depressed (%) (Weighted; n=9396)	Depressed (%) (Weighted; n=604)	Total		
			Proportion (Weighted) ₁	N (%) (Not weighted)	Proportion ²
Sex					
Female	49.1%	46.2%	48.9%	4,921 (49.2%)	48.9%
Age Group					
< 20 Years	10.5%	11.0%	10.5%	900 (9.0%)	6.9%
20-29 Years	17.1%	17.9%	17.1%	1,645 (16.5%)	20.8%
30-39 Years	18.1%	15.7%	17.9%	1,895 (19.0%)	18.2%
40-49 Years	18.7%	20.5%	18.9%	1,890 (18.9%)	22.1%
50-59 Years	17.9%	20.1%	18.0%	1,820 (18.2%)	16.5%
≥ 60 Years	17.7%	14.9%	17.5%	1,675 (16.8%)	15.4%
Highest Education Level					
less than high school	24.8%	25.5%	24.8%	2792 (27.0%)	68.5%
High School / Technical Secondary School	37.1%	36.8%	37.1%	3,733 (37.3%)	17.6%

College / Undergraduate	36.0%	35.7%	36.0%	3,369 (33.7%)	13.4 %
Graduate school and above	2.1%	2.0%	2.1%	206 (2.0%)	0.6 %
Ethnicity					
Han	95.1%	94.8%	95.1%	9,381 (93.8%)	95.0 %
Man	0.4%	0.6%	0.5%	149 (1.5%)	0.7 %
Hui	0.1%	0.2%	0.1%	109 (1.1%)	0.8 %
Zang	1.8%	1.5%	1.8%	103 (1.0%)	0.5 %
Zhuang	1.4%	1.5%	1.4%	152 (1.5%)	1.2 %
Other	1.1%	1.5%	1.1%	106 (1.1%)	1.8 %
Province of Residence					
Anhui	4.6%	4.0%	4.5%	360 (3.6%)	4.5 %
Beijing	1.5%	1.9%	1.5%	360 (3.6%)	1.5 %
Chongqing	2.2%	2.4%	2.2%	360 (3.6%)	2.2 %
Fujian	2.8%	2.9%	2.8%	300 (3.0%)	2.8 %
Gansu	1.9%	1.8%	1.9%	300 (3.0%)	1.9 %
Guangdong	8.1%	9.5%	8.2%	360 (3.6%)	8.2 %
Guangxi	3.5%	3.8%	3.5%	300 (3.0%)	3.5 %
Guizhou	2.6%	1.8%	2.6%	300 (3.0%)	2.6 %
Hainan	0.7%	0.4%	0.7%	300 (3.0%)	0.7 %

Hebei	5.4%	5.5%	5.4%	360 (3.6%)	5.4%
Heilongjiang	2.7%	2.1%	2.7%	300 (3.0%)	2.7%
Henan	6.6%	10.3%	6.9%	360 (3.6%)	6.9%
Hubei	4.2%	4.3%	4.2%	360 (3.6%)	4.2%
Hunan	4.9%	4.7%	4.9%	300 (3.0%)	4.9%
Jiangsu	5.7%	7.1%	5.7%	360 (3.6%)	5.7%
Jiangxi	3.4%	2.8%	3.3%	300 (3.0%)	3.3%
Jilin	1.9%	1.8%	1.9%	300 (3.0%)	1.9%
Liaoning	3.1%	2.6%	3.1%	340 (3.4%)	3.1%
Neimengol	1.9%	1.2%	1.8%	300 (3.0%)	1.8%
Ningxia	0.5%	0.4%	0.5%	300 (3.0%)	0.5%
Qinghai	0.4%	0.3%	0.4%	300 (3.0%)	0.4%
Shaanxi	2.8%	2.6%	2.7%	360 (3.6%)	2.7%
Shandong	7.3%	5.7%	7.2%	360 (3.6%)	7.2%
Shanghai	1.7%	1.4%	1.7%	300 (3.0%)	1.7%
Shanxi	2.7%	1.7%	2.8%	300 (3.0%)	2.8%
Sichuan	5.9%	7.1%	6.0%	360 (3.6%)	6.0%
Tianjin	1.1%	1.0%	1.1%	360 (3.6%)	1.1%
Tibet	0.2%	0.3%	0.3%	300 (3.0%)	0.3%

Xinjiang	1.8%	2.2%	1.8%	300 (3.0%)	1.8%
Yunnan	3.5%	2.7%	3.5%	300 (3.0%)	3.5%
Zhejiang	4.2%	3.5%	4.2%	360 (3.6%)	4.2%
Residence Type					
Urban	60.9%	62.9%	61.1%	5,935 (59.4%)	60.6%
Health Care Providers					
No	96.2%	93.6%	96.0%	9,597 (96.0%)	99.0%
Nurse	0.4%	1.3%	0.5%	55 (0.6%)	0.3%
Physician	0.8%	1.3%	0.9%	84 (0.8%)	0.5%
Community Health Worker	1.4%	2.5%	1.5%	157 (1.6%)	<0.1%
Pharmacist	0.1%	0.1%	0.1%	17 (0.2)	<0.1%
Other Healthcare Provider	0.9%	1.2%	1.0%	90 (0.9%)	0.1%
Annual Household Income (RMB)					
< 30,000	5.7%	6.5%	5.8%	560 (5.6%)	
30,000 - 59,999	15.0%	13.9%	14.9%	1,670 (16.7%)	-
60,000 - 89,999	21.7%	19.6%	21.6%	2,303 (23.0%)	-
90,000 - 119,999	25.9%	30.9%	26.2%	2,704 (24.0%)	-
120,000 - 149,999	14.2%	14.9%	14.2%	1,211 (12.1%)	-
150,000 - 199,999	11.1%	8.8%	11.1%	974 (9.7%)	-
≥ 200,000	6.4%	5.4%	6.4%	578(5.8%)	-

Personal COVID-19 Diagnosis History				
No	99.9%	99.8%	99.9%	9,992 (99.9%)
Yes	0.1%	0.2%	0.1%	8(0.08%)
Positive Acquaintance COVID-19 Diagnosis				
Family member	0.1%	0.2%	0.1%	5(0.05%)
Friend	0.2%	0.0%	0.1%	12(0.12%)
Neighbor	0.1%	0.0%	0.1%	7(0.07%)
Cowoker	0.1%	0.3%	0.1%	10(0.1%)
Other	0.0%	0.0%	0.0%	1(0)
COVID-19 Awareness				
Perceived risk of death among vulnerable groups	3.3% (IQR: 1.0%-4.0%)	3.5% (IQR: 1.0%-4.0%)	3.3% (IQR: 1.0%-4.0%)	3.3% (IQR: 1.0%-4.0%)
Perceived risk of death among people with other diseases	96.6% (95% CI: 96.2% - 96.9%)	94.4% (95% CI: 92.2% - 96.0%)	96.4% (95% CI: 96.0% - 96.8%)	96.3% (95% CI: 96.0% - 96.7%)
Perceived elderly as a high-risk group of transmission	92.6% (95% CI: 92.0% - 93.1%)	92.6% (95% CI: 90.1% - 94.4%)	92.6% (95% CI: 92.0% - 93.1%)	92.7% (95% CI: 92.2% - 93.2%)
Awareness of vaccine availability protecting against transmission	76.7% (95% CI: 75.8% - 77.5%)	74.9% (95% CI: 71.2% - 78.1%)	76.5% (95% CI: 75.7% - 77.3%)	76.0% (95% CI: 75.1% - 76.8%)
Awareness that masks are highly effective in protecting against transmission	89.3% (95% CI: 88.7% - 89.9%)	88.0% (95% CI: 85.1% - 90.4%)	89.2% (95% CI: 88.6% - 89.8%)	89.8% (95% CI: 89.2% - 90.4%)
Knowledge of transmission prevention actions	11.6% (95% CI: 10.9% - 12.2%)	8.1% (95% CI: 6.2% - 10.6%)	11.4% (95% CI: 10.8% - 12.0%)	11.1% (95% CI: 10.5% - 11.7%)
Knowledge of saliva as main transmission route	82.6% (95% CI: 81.8% - 83.3%)	76.9% (95% CI: 73.4% - 80.1%)	82.2% (95% CI: 81.5% - 83.0%)	82.1% (95% CI: 81.3% - 82.8%)
Awareness of COVID-19 symptoms	69.4% (95% CI: 68.5% - 70.3%)	64.4% (95% CI: 60.5% - 68.1%)	69.1% (95% CI: 68.2% - 70.0%)	67.1% (95% CI: 66.1% - 68.0%)

Awareness of recommended healthcare-seeking behavior	71.6% (95% CI: 70.7% - 72.5%)	68.2% (95% CI: 64.3% - 71.8%)	71.4% (95% CI: 70.5% - 72.3%)	68.8% (95% CI: 67.9% - 69.7%)
---	-------------------------------	-------------------------------	-------------------------------	-------------------------------

¹ Weighted using survey sampling weights.

² As per the 2020 China Statistical Yearbook.

For dichotomous outcomes, data are expressed as a percentage with the correct response (95% confidence interval). For continuous outcomes, data are expressed as median (interquartile range).

Figure 1. Proportion of the population reporting depression (PHQ-9 ≥ 10) by province* (10,000 participants, May 8-June 8, 2020)

* Taiwan, Hong Kong and Macao are shown in grey.

Source of map: http://datav.aliyun.com/portal/school/atlas/area_selector ”

[Comment-10]

10) *The PHQ-9 cut-off described in the method was ≥ 4 and this was reported as greater than 4 in the discussion. Please provide the correct description.*

[Answer] Thanks for this comment. The authors corrected the description about the PHQ-9 cut-off in the discussion section. **(Page 14, paragraph 1)**

“In our online survey sample, 6.3% of adults had depression as defined by a PHQ-9 ≥ 10 .”

[Comment-11]

11) Discussion of depression rates identified in other studies needs to be better contextualised. For instance, are there other nationally representative studies that have used the PHQ-9 during COVID-19 and reported on the same cut-offs? The rate of depression identified will depend on the measure and cut-off used which needs to be noted.

[Answer] Thanks for this comment. The authors added the following text to the Discussion section. **(Page 14, paragraph 3)**

“Several studies have described the prevalence of depression during the COVID-19 pandemic in the general population. Among studies using the PHQ-9 scale with the same cut-off value (≥ 10), the prevalence of depression observed in our study (6.3%) was lower than a national study among 56,679 participants conducted February 28 to March 11, 2020 in China (10.8%)²⁹ as well as a study of 1,470 individuals among the general population during the COVID-19 outbreak in the United States from March 31 to April 13, 2020 (27.8%)⁴²⁻⁴³. Most of the other studies were conducted in February 2020—at the peak of the COVID-19 epidemic, which ranged from 8.3 to 48.3%²⁹⁻³¹⁻³³⁻⁴⁴⁻⁴⁶. One possible explanation is that our study occurred in May when the epidemic had decreased in severity, and the prevalence of depression may have diminished. Another possible reason for a lower depression prevalence found in our study may be that with the continuous strict quarantine policy, the government provided timely mental health service in response to COVID-19³⁴.”

Response to Reviewer 3

[Comment-1] 1. Page 7, line 20, “yen” → “yuan”.

[Answer] Thank you for this comment. The spelling has been corrected in the Methods section. **(Page 7, paragraph 1)**

“Participants were paid 5 yuan (US\$0.77) for completing the questionnaire.”

[Comment-2] 2. Page 7, line 48, “a score ≥ 4 ”. Why do we use 4 as the threshold? Is it for a more balanced data in logistic regression? Or do we have literature convention or argument from mental health studies? In some other places of the paper, the denition becomes > 4 (e.g. Page 9, line 48). In Table 1, no depression is defined as $PHQ-9 \leq 4$ and depression is defined as $PHQ-9 \geq 4$. Please be consistent with how to handle the case with $PHQ-9 = 4$.

[Answer] Thank you for this comment. The authors have revised the cut-off to ≥ 10 as the depression symptom definition and updated Table 1 and Table 2 as well **(Page 10-13)**. The results and discussion sections were also updated to reflect this change. **(Page 9-17)**. The PHQ-9 cut-off was revised in the methods **(Page 8, paragraph 1)** and discussion section **(Page 14, paragraph 1)**.

“We defined a score of 10 or greater as depression;”

“In our online survey sample, 6.3% of adults had depression as defined by a PHQ-9 ≥ 10 .”

[Comment-3]3. Page 8, line 36, “using 2 to compare”. I do not follow what the authors mean here. Please elaborate.

[Answer] Thank you for this comment. The authors have rephased this sentence for clarity as suggested. (Page 8, paragraph 2)

“The prevalence of depression was stratified by participants’ sociodemographic characteristics and overall knowledge about COVID-19.”

[Comment-4]4. Page 8, line 43, “an odds ratio” → “odds ratios”.

[Answer] Thank you for this suggestion. The authors have modified the following text in the methods section. (Page 8, paragraph 2)

“We used covariate-unadjusted and covariate-adjusted logistic regression with a binary indicator for each province (province-level fixed effects) and obtained odds ratios (OR).”

[Comment-5]5. The paper has no clear reference to Table 1. Please add a sentence in the main text about something like “Please refer to Table 1 on summary statistics of ...”.

[Answer] The authors added the following sentence in the results section. (Page 9, paragraph 3)

“...(See Table 1).”

[Comment-6] 6. Page 15, lines 27-34. I do not follow the sentence. That the prevalence of depression had not been alleviated may explain why the prevalence was still high. How can we explain A by A itself?

[Answer] Thank you for this question. Since depression is defined as PHQ-9 ≥ 10 , the following text has been added to the Discussion section. (Page 14, paragraph 3)

“Several studies have described the prevalence of depression during the COVID-19 pandemic in the general population. Among studies using the PHQ-9 scale with the same cut-off value (≥ 10), the prevalence of depression observed in our study (6.3%) was lower than a national study among 56,679 participants conducted February 28 to March 11, 2020 in China (10.8%)²⁹ as well as a study of 1,470 individuals among the general population during the COVID-19 outbreak in the United States from March 31 to April 13, 2020 (27.8%)^{42 43}. Most of the other studies were conducted in February 2020—at the peak of the COVID-19 epidemic, which ranged from 8.3 to 48.3%^{29 31-33 44-46}. One possible explanation is that our study occurred in May when the epidemic had decreased in severity, and the prevalence of depression may have diminished. Another possible reason for a lower depression prevalence found in our study may be that with the continuous strict quarantine policy, the government provided timely mental health service in response to COVID-19³⁴.”

[Comment-7] 7. Page 17, line 6, “a effective communiatiom” → “effective communication”

[Answer] Thank you for this suggestion. The authors have made this change as suggested. (Page 16, paragraph 2)

“This finding suggests that, to reduce the prevalence of depression, effective communication and education of COVID-19 preventive measures and recommended healthcare-seeking behaviors are urgently needed.”

[Comment-8] 8. Page 17, lines 13-22. People in urban areas have higher education and greater access.”to COVID information. How come their depression is higher or positively correlated with urban dummy variable? The authors argued more knowledge on COVID should reduce the depression. They seem to contradict here. Is it possible that the higher depression may not be explained as something related with COVID, but just by the fact that working in a city is more stressful?

[Answer] Thank you for this comment, the authors agree. The previous wording may have been unclear. We improved the following text in the discussion section. **(Page 14, paragraph 2)**

“A possible interpretation of this finding is that while the virus could be transmitted more quickly in urban areas with a higher density population²⁸, those in urban areas tend to have higher education levels and greater access to the latest report on the COVID-19 pandemic²⁶. Another explanation is that depression was more common in the urban than in rural areas in China before the COVID-19 pandemic.”

[Comment-9] 9. Page 17, line 31, “more vulnerable” → “are more vulnerable”.
10. Page 17, line 34, “contracting” → “contacting”.

[Answer] Thank you for these suggestions. Since depression is defined in the revised manuscript as PHQ-9 ≥ 10, the results have been revised. No association was found between those with a family member or neighbor with confirmed COVID-19 and depression. Accordingly these sentences have been removed.

[Comment-10] 11. Page 17, line 45, “initial stage”? The data was collected in May 2020. Is this still considered as initial stage? By this time, the peak outbreak of the pandemic in China has passed.

[Answer] The authors improved the following text in the limitations section. **(Page 16, paragraph 3)**

“This is the first study that investigated the prevalence of depression by province during the early stage of the COVID-19 pandemic in China;”

[Comment-11] 12. Page 18, lines 4-6. The study design was cross sectional, so the authors “cannot make temporal conclusions”. But the paper does compare the prevalence of depression with previous studies in lines 10-18, Page 15. Could you explain what the meaning of “temporal conclusions” is here?

[Answer] Thank you for your comment. This phrasing has been removed as suggested.

[Comment-12] 13. Page 18, line 6, “connont” → “cannot”.

14. Page 18, line 18, “was associated” → “were associated”.

[Answer] Thank you for these corrections. The authors have improved the following text in the Limitations section. **(Page 17, paragraph 1 and Page 17 paragraph 2)**

“Although we used stratified sampling to increase the representativeness of the data, it is still difficult to avoid response bias as potential participants with depression might be either less or more interested in taking part in the survey.”

Accurate knowledge of COVID-19 transmission and awareness of COVID-19 symptoms were associated with lower odds of depression.”

Reference

1. National Bureau of Statistics of China. The urban-rural composition of the population by region and the birth rate, death rate, and natural growth rate (2019) Beijing2020 [cited 2021 May 30]. Available from: <http://www.stats.gov.cn/tjsj/ndsj/2020/indexch.htm> accessed May 30 2021.
2. World Health Organization. Coronavirus disease (COVID-19) pandemic [Available from: https://covid19.who.int/?gclid=Cj0KCQjw8IaGBhCHARIsAGIRRYpDW4NS0CUiiW6a4fYWJ-AdgKZk4MDgbs9ck5gmi1c21JXMESfvjZwaAkP4EALw_wcB accessed 18 January 2022.
3. Fann JR, Berry DL, Wolpin S, et al. Depression screening using the Patient Health Questionnaire-9 administered on a touch screen computer. *Psycho-oncology* 2009;18(1):14-22. doi: 10.1002/pon.1368
4. Spitzer RL, Kroenke K, Williams JB. Validation and utility of a self-report version of PRIME-MD: the PHQ primary care study. Primary Care Evaluation of Mental Disorders. Patient Health Questionnaire. *Jama* 1999;282(18):1737-44. doi: 10.1001/jama.282.18.1737
5. Kroenke K, Spitzer RL, Williams JB. The PHQ-9: validity of a brief depression severity measure. *J Gen Intern Med* 2001;16(9):606-13. doi: 10.1046/j.1525-1497.2001.016009606.x
6. Wang W, Bian Q, Zhao Y, et al. Reliability and validity of the Chinese version of the Patient Health Questionnaire (PHQ-9) in the general population. *General hospital psychiatry* 2014;36(5):539-44. doi: 10.1016/j.genhosppsy.2014.05.021
7. Zhang YL, Liang W, Chen ZM, et al. Validity and reliability of Patient Health Questionnaire-9 and Patient Health Questionnaire-2 to screen for depression among college students in China. *Asia-Pacific psychiatry : official journal of the Pacific Rim College of Psychiatrists* 2013;5(4):268-75. doi: 10.1111/appy.12103
8. Kroenke K, Spitzer RL. The PHQ-9: A New Depression Diagnostic and Severity Measure. *Psychiatr Ann* 2002;32(9):509-21.
9. Fann JR, Berry DL, Wolpin S, et al. Depression screening using the Patient Health Questionnaire-9 administered on a touch screen computer. *Psycho-Oncology: Journal of the Psychological, Social Behavioral Dimensions of Cancer* 2009;18(1):14-22.
10. Spitzer RL, Kroenke K, Williams JB, et al. Validation and utility of a self-report version of PRIME-MD: the PHQ primary care study. *Jama* 1999;282(18):1737-44.
11. Kroenke K, Spitzer RL, Williams JB. The PHQ-9: validity of a brief depression severity measure. *Journal of general internal medicine* 2001;16(9):606-13.
12. Wang W, Bian Q, Zhao Y, et al. Reliability and validity of the Chinese version of the Patient Health Questionnaire (PHQ-9) in the general population. *General hospital psychiatry* 2014;36(5):539-44.
13. Zhang Y, Liang W, Chen Z, et al. Validity and reliability of Patient Health Questionnaire-9 and Patient Health Questionnaire-2 to screen for depression among college students in China. *Asia-Pacific Psychiatry* 2013;5(4):268-75.
14. Yu F, Geldsetzer P, Meierkord A, et al. Knowledge About COVID-19 Among Adults in China: Cross-sectional Online Survey. *Journal of Medical Internet Research* 2021;23(4):e26940.
15. Cabarkapa S, Nadjidai SE, Murgier J, et al. The psychological impact of COVID-19 and other viral epidemics on frontline healthcare workers and ways to address it: A rapid systematic review. *Brain, behavior, immunity-health* 2020:100144.
16. Sriharan A, West KJ, Almost J, et al. COVID-19-Related Occupational Burnout and Moral Distress among Nurses: A Rapid Scoping Review. *Nursing leadership (Toronto, Ont)* 2021;34(1):7-19. doi: 10.12927/cjnl.2021.26459
17. Lai J, Ma S, Wang Y, et al. Factors associated with mental health outcomes among health care workers exposed to coronavirus disease 2019. *JAMA network open* 2020;3(3):e203976-e76.
18. Bareaq SB, Ahmed SI, Samar SS, et al. Prevalence of depression, anxiety and stress in china during COVID-19 pandemic: A systematic review with meta-analysis. *International journal of psychiatry in medicine* 2021;56(4):210-27. doi: 10.1177/0091217420978005
19. Huang Y, Zhao N. Generalized anxiety disorder, depressive symptoms and sleep quality during COVID-19 outbreak in China: a web-based cross-sectional survey. *Psychiatry Res* 2020;288:112954. doi: 10.1016/j.psychres.2020.112954
20. Wang C, Pan R, Wan X, et al. A longitudinal study on the mental health of general population during the COVID-19 epidemic in China. *Brain, behavior, and immunity* 2020;87:40-48. doi: 10.1016/j.bbi.2020.04.028

21. Qiu J, Shen B, Zhao M, et al. A nationwide survey of psychological distress among Chinese people in the COVID-19 epidemic: implications and policy recommendations. *Gen Psychiatr* 2020;33(2):e100213. doi: 10.1136/gpsych-2020-100213
22. Wang C, Pan R, Wan X, et al. Immediate Psychological Responses and Associated Factors during the Initial Stage of the 2019 Coronavirus Disease (COVID-19) Epidemic among the General Population in China. *Int J Environ Res Public Health* 2020;17(5) doi: 10.3390/ijerph17051729
23. World Health Organization. The COVID-19 vaccine tracker and landscape compiles detailed information of each COVID-19 vaccine candidate in development by closely monitoring their progress through the pipeline 2022 [Available from: <https://www.who.int/publications/m/item/draft-landscape-of-covid-19-candidate-vaccines>.
24. Hannah Ritchie EM, Lucas Rodés-Guirao, Cameron Appel, Charlie Giattino, Esteban Ortiz-Ospina, Joe Hasell, Bobbie Macdonald, Diana Beltekian and Max Roser. Coronavirus Pandemic (COVID-19): Our World in Data; 2022 [Available from: <https://ourworldindata.org/coronavirus>.
25. Tracker OC-GR. Government Response Stringency Index. *Our World in Data*
26. Özdin S, Bayrak Özdin Ş. Levels and predictors of anxiety, depression and health anxiety during COVID-19 pandemic in Turkish society: The importance of gender. *International Journal of Social Psychiatry* 2020;66(5):504-11.
27. Chen Y, Jin Y, Zhu L, et al. The network investigation on knowledge, attitude and practice about COVID-19 of the residents in Anhui Province. *Zhonghua yu fang yi xue za zhi* 2020;54(4):367-73.
28. Taylor S. The psychology of pandemics: Preparing for the next global outbreak of infectious disease: Cambridge Scholars Publishing 2019.
29. Gao J, Zheng P, Jia Y, et al. Mental health problems and social media exposure during COVID-19 outbreak. *Plos one* 2020;15(4):e0231924.
30. Shi L, Lu Z, Que J, et al. Prevalence of and risk factors associated with mental health symptoms among the general population in China during the coronavirus disease 2019 pandemic. *JAMA network open* 2020;3(7):e2014053-e53.
31. Qiu J, Shen B, Zhao M, et al. A nationwide survey of psychological distress among Chinese people in the COVID-19 epidemic: implications and policy recommendations. *General psychiatry* 2020;33(2)
32. Ahmed MZ, Ahmed O, Aibao Z, et al. Epidemic of COVID-19 in China and associated psychological problems. *Asian journal of psychiatry* 2020;51:102092.
33. Zhou S, Zhang L, Wang L, et al. Prevalence and socio-demographic correlates of psychological health problems in Chinese adolescents during the outbreak of COVID-19. *European Child Adolescent Psychiatry* 2020;29(6):749-58.
34. Ju Y, Zhang Y, Wang X, et al. China's mental health support in response to COVID-19: progression, challenges and reflection. *Globalization and Health* 2020;16(1):102. doi: 10.1186/s12992-020-00634-8
35. Bitsika V, Sharpley CF, Bell R. The buffering effect of resilience upon stress, anxiety and depression in parents of a child with an autism spectrum disorder. *Journal of Developmental Physical Disabilities* 2013;25(5):533-43.
36. Chen S, Yang J, Yang W, et al. COVID-19 control in China during mass population movements at New Year. *The Lancet* 2020;395(10226):764-66.
37. Chen S, Chen Q, Yang J, et al. Curbing the COVID-19 pandemic with facility-based isolation of mild cases: a mathematical modeling study. *Journal of Travel Medicine* 2020;28(2) doi: 10.1093/jtm/taaa226
38. Dickens BL, Koo JR, Wilder-Smith A, et al. Institutional, not home-based, isolation could contain the COVID-19 outbreak. *The Lancet* 2020;395(10236):1541-42.
39. Chen S, Zhang Z, Yang J, et al. Fangcang shelter hospitals: a novel concept for responding to public health emergencies. *The Lancet* 2020;395(10232):1305-14.
40. Qian Y, Wu K, Xu H, et al. A survey on physical and mental distress among cancer patients during the COVID-19 epidemic in Wuhan, China. *Journal of palliative medicine* 2020;23(7):888-89.
41. Hammen CL. Stress and depression: old questions, new approaches. *Current Opinion in Psychology* 2015;4:80-85.
42. Shi L, Lu Z-A, Que J-Y, et al. Prevalence of and risk factors associated with mental health symptoms among the general population in China during the coronavirus disease 2019 pandemic. *JAMA network open* 2020;3(7):e2014053-e53.

43. Shi L, Lu ZA, Que JY, et al. Prevalence of and Risk Factors Associated With Mental Health Symptoms Among the General Population in China During the Coronavirus Disease 2019 Pandemic. *JAMA Netw Open* 2020;3(7):e2014053. doi: 10.1001/jamanetworkopen.2020.14053
44. Huang Y, Zhao N. Generalized anxiety disorder, depressive symptoms and sleep quality during COVID-19 outbreak in China: a web-based cross-sectional survey. *Psychiatry research* 2020;288:112954.
45. Wang C, Pan R, Wan X, et al. A longitudinal study on the mental health of general population during the COVID-19 epidemic in China. *Brain, behavior*, 2020;87:40-48.
46. Wang C, Pan R, Wan X, et al. Immediate psychological responses and associated factors during the initial stage of the 2019 coronavirus disease (COVID-19) epidemic among the general population in China. *International journal of environmental research public health* 2020;17(5):1729.

VERSION 2 – REVIEW

REVIEWER	Daly, Michael National University of Ireland
REVIEW RETURNED	09-Feb-2022
GENERAL COMMENTS	The paper has been revised extensively and comments addressed thoroughly. I have no further comments.